# Multi-platform profiling characterizes molecular subgroups and resistance networks in chronic lymphocytic leukemia

Johannes Bloehdorn [1✉], Andrejs Braun [2], Amaro Taylor-Weiner[3], Billy Michael Chelliah Jebaraj[1], Sandra Robrecht[4], Julia Krzykalla[5], Heng Pan[6,7,8], Adam Giza[4], Gulnara Akylzhanova[2], Karlheinz Holzmann[9], Annika Scheffold[1], Harvey E. Johnston[10], Ru-Fang Yeh[11], Tetyana Klymenko[2], Eugen Tausch[1], Barbara Eichhorst[4], Lars Bullinger[12], Kirsten Fischer[4], Martin Weisser[13], Tadeusz Robak [14], Christof Schneider[1], John Gribben [2], Lekh N. Dahal [10,15], Mathew J. Carter[10], Olivier Elemento [6,7,8,16], Dan A. Landau [17,18], Donna S. Neuberg [19], Mark S. Cragg[10], Axel Benner [5], Michael Hallek[4], Catherine J. Wu [3,20,21,22], Hartmut Döhner[1], Stephan Stilgenbauer[1] & Daniel Mertens[1,23✉]

Knowledge of the genomic landscape of chronic lymphocytic leukemia (CLL) grows increasingly detailed, providing challenges in contextualizing the accumulated information. To define the underlying networks, we here perform a multi-platform molecular characterization. We identify major subgroups characterized by genomic instability (GI) or activation of epithelial-mesenchymal-transition (EMT)-like programs, which subdivide into non-inflammatory and inflammatory subtypes. GI CLL exhibit disruption of genome integrity, DNA-damage response and are associated with mutagenesis mediated through activation-induced cytidine deaminase or defective mismatch repair. *TP53* wild-type and mutated/deleted cases constitute a transcriptionally uniform entity in GI CLL and show similarly poor progression-free survival at relapse. EMT-like CLL exhibit high genomic stability, reduced benefit from the addition of rituximab and EMT-like differentiation is inhibited by induction of DNA damage. This work extends the perspective on CLL biology and risk categories in *TP53* wild-type CLL. Furthermore, molecular targets identified within each subgroup provide opportunities for new treatment approaches.

A full list of author affiliations appears at the end of the paper.

Characterization of genetic heterogeneity and its related clinical impact has provided the fundament for prognostic models in chronic lymphocytic leukemia (CLL) and has been extended considerably in recent years[1–3]. However, the context in which genetic alterations arise remains to be further explored to understand disease dynamics and to refine therapeutic strategies by targeting cellular networks or genetic dependencies. Alterations of the tumor suppressor genes *TP53* and *ATM* have been identified as major determinants for dysfunctional DNA-damage response (DDR), genomic instability, selection of genomically complex clones, and poor response to treatment[3–8]. Treatment with genotoxic substances was found to contribute to the inactivation of these tumor suppressors, acquisition of chromosomal aberrations, and clonal evolution[7–10]. However, the mechanisms inducing genomic instability in cases without such lesions are incompletely characterized. Correspondingly, it remains to be proven if genomic lesions occur randomly or are specifically selected within a defined molecular or biologic framework during malignant transformation and over the natural course of CLL. Other treatment-independent factors contributing to the selection of alterations may include a heterogeneous degree of addiction to environmental stimuli or necessity to maintain the activity of certain signaling pathways[11,12]. Further, genomic lesions may evolve in a narrow spectrum depending on the related epigenetic makeup[13,14].

In this work, we aim to delineate refined biological categories of CLL and identify cooperating pathogenic mechanisms which facilitate distinct pathways or microenvironmental interaction during disease development and evolution. We address this by performing a comprehensive characterization incorporating gene expression profiles (GEP) from two independent phases III CLL trial cohorts comprising 726 treatment-naive and relapsed patient samples. Data from whole-exome sequencing (WES), single-nucleotide polymorphism (SNP)-array analysis, and protein expression is included for detailed biological characterization of the discovery cohort containing samples from untreated CLL patients enrolled on the CLL8 trial[15]. Discovered biologic subgroups are validated in the independent sample set of relapsed patients enrolled onto the REACH trial[16] and then confirmed in vivo using relevant genetically modified mouse models. Both the CLL8 and REACH trials were conducted as independent pivotal phase III multicenter trials to evaluate treatment with immunochemotherapy[15,16]. They, therefore, provide an ideal basis for the correlation of biological characteristics and treatment outcome.

## Results

### Consensus clustering identifies distinct expression signatures associated with inflammation, genomic instability, and activation of EMT-like networks.

To explore tumor heterogeneity in CLL, we performed consensus clustering (CC) on CLL8 GEP data ($n = 337$, Supplementary Table 1) using 2359 variably expressed genes corresponding to a standard deviation (SD) of >0.5. First, the two initial clusters identified for $k = 2$ and building the most distant branches of the dendrogram (protocluster C1 and C2) (Supplementary Fig. 1a) were assessed for discriminatory characteristics. Using gene set enrichment analysis (GSEA), inflammatory features were identified as most prominent in segregating protocluster C1 and C2 and the subsequently derived clusters. To further decipher the underlying biology, we performed CC with an increasing number of clusters (Supplementary Fig. 1a–d) and serial analytical steps by incorporating additional layers of information as schematically shown (Fig. 1a, Supplementary Fig. 1e). Optimal differentiation of distinct subtypes was achieved for $k = 6$ GEP clusters (Fig. 1b, Supplementary Fig. 1e, f), while DNA-based class discovery approaches were insufficient to uncover similar patterns and the

respective biological context (Supplementary Fig. 1g). Subtypes were labeled according to the most prominent characteristics (Fig. 1a) as "genomically instable, non-inflammatory" (GI) for C2 ($n = 133$), "genomically instable with inflammatory features" ((I)GI) for C3 ($n = 56$), "epithelial–mesenchymal-transition-like, non-inflammatory" (EMT-L) for C4 ($n = 30$) and "epithelial-mesenchymal-transition-like with inflammatory features" ((I)EMT-L) for C1 ($n = 100$). C5 cases ($n = 11$) were labeled as "reprogrammed by early B cell factor 1" (EBF1-r), identifying tri(12) CLL as a distinct subtype with strong overexpression of *EBF1* and a transcriptional signature resembling healthy B cells. C6 ($n = 7$) identified nuclear receptor-interacting protein 1 (*NRIP1*) as specific for clusters evolving from the protocluster C1 (Supplementary Fig. 2a). *NRIP1* is associated with clinical outcomes in CLL and is a major regulator of metabolism and coactivator of NF-kB-dependent inflammation [17].

Cases with *TP53* inactivation (Fig. 1b, Supplementary Table 2), V3-21 usage (Supplementary Table 3), short telomeres (Fig. 1c), high white blood cell (WBC) counts (Fig. 1d, Supplementary Table 2), and ZAP-70 positivity (Supplementary Table 3) ($p < 0.05$, Fisher's exact test (two-sided)) were enriched in GI/(I)GI clusters. *TP53* mutated cases without concomitant del(17p) showed a near-exclusive occurrence in genomically instable cases (GI/(I)GI: $n = 16$ (9.5%) vs. EMT-L/(I)EMT-L: $n = 1$ (0.8%)) ($p = 0.002$, Fisher's exact test (two-sided)). GSEA identified processes associated with genomic instability for GI/(I)GI and EMT networks for EMT-L/(I)EMT-L (Fig. 2a). Furthermore, genes involved in the maintenance of genomic stability were frequently mutated (Supplementary Fig. 2b, c) or overexpressed (Supplementary Fig. 2d) in the GI/(I)GI cluster.

In summary, CLL can be segregated into two major biological subgroups defined by genomic instability or EMT-like networks with variable degrees of inflammatory features, further characterizing the inflammatory or non-inflammatory subtypes. GI and (I) EMT-L comprise the two largest and most distinct subtypes. Of importance, transcriptional homogeneity and consecutive co-clustering of CLL with GI/(I)GI expression signatures and cases showing *TP53* inactivation indicate that changes in genes other than *TP53* may execute similar biological effects and contribute to genomic instability.

### Frequency and distribution of copy number alterations support a higher susceptibility to DNA damage in genomically instable CLL.

To further investigate genomic instability, we assessed the distribution of copy number alterations (CNAs) using SNP-array. Genomic identification of significant targets in cancer (GISTIC)[18] was used to identify significantly altered regions and candidate genes representing putative targets of focal chromosomal amplification or loss (Fig. 2b). Both GI and (I)GI involved frequent gains of 8q24.21 (including *MYC*) and 2p16.1 (including *XPO1*, *REL*). GI further showed gains of 6q22.31 and losses of 15q15.1 (including *KNSTRN*, *BUB1B*), 10q24.3, and 6q21. (I)GI showed losses of 13q14.13. (I)EMT-L had losses for 6q21 and 14q32.1 (Fig. 2b). Although 11q deletions were found in all clusters at a similar frequency, genes covered on 11q22.1–q22.2 (involving *YAP1* and an *MMP* cluster) were predicted by GISTIC to be specifically lost in GI (Supplementary Fig. 2e–g) and showed a confirmatory underexpression (Supplementary Fig. 2h). Assessing the impact of CNAs on expression, we found cluster-specific profiles of differentially expressed genes (DEGs) located in and adjacent to the minimally deleted or minimally gained regions, irrespective of the distribution of single CNAs (Fig. 2c). This indicates that gene dosage effects for monoallelic deletions are context-dependent and modulate dominant pathogenic networks. Context-independent gene dosage effects were only observed in cases with biallelic deletion

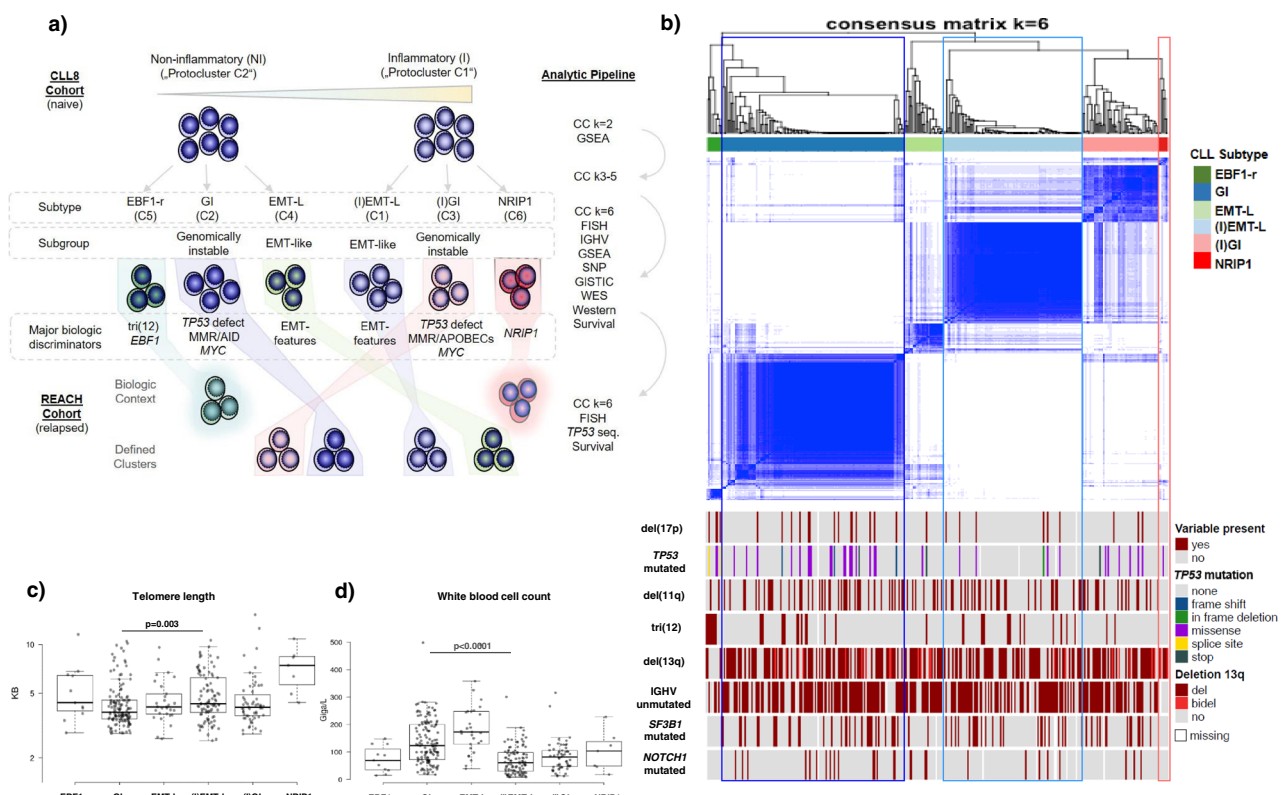

**Fig. 1 Composition and relationship of CLL subtypes in clustered data. a** Schematic representation for analysis, identification of CLL subtypes in the CLL8, and confirmation in the REACH cohort. The four largest clusters (GI, (I)GI, EMT-L, (I)EMT-L), and associations of *NRIP1* with the inflammatory or tri(12) with the EBF1-r signature were also identified in the independent validation cohort of the REACH trial. Co-clustering of GI/(I)GI and EMT-L/(I)EMT-L cases in the REACH cohort supports the selection of subgroup-specific characteristics during treatment. **b** Heatmap showing the consensus clustering for $k = 6$ used for defining CLL subtypes ($n = 337$). The distribution of genetic characteristics is shown below the heatmap. Significant enrichment of variables in clusters is observed for del(17p) ($p = 0.05$), *TP53* mutation ($p = 0.01$), tri(12) ($p = 7e-06$), del(13q) ($p = 0.03$), and IGHV mutation status ($p = 0.008$) (all Fisher's exact test (two-sided)). *TP53* frameshift mutations occur exclusively in GI and splice site mutations in EBF1-r cases. Tri(12) is strongly overrepresented in EBF1-r (72.7%). **c** Telomere length is significantly different across CLL subtypes ($p < 0.001$, Kruskal–Wallis test), and shortest length is observed in GI with a median of 3.8 kb ($p = 0.003$, Mann–Whitney (two-sided), for GI vs. (I)EMT-L) ($n = 333$). **d** White blood cell counts are significantly different across CLL subtypes (p < 0.0001, Kruskal-Wallis test), show decreased counts in inflammatory CLL and are lowest in (I)EMT-L with median 61.1 G/L ($p < 0.0001$, Mann–Whitney (two-sided), for GI vs. (I)EMT-L) ($n = 330$). For Fig. 1a–d, data within individual figures derives from biologically independent samples. For the boxplots, centerline, box limits, and whiskers represent the median, 25th, and 75th percentiles, and 1.5× interquartile range, respectively.

of 13q14 (Fig. 2d). Significant enrichment and co-occurrence of deletions involving *RB1* (previously defined as type II deletions)[19] and losses exceeding cytoband 13q21.1, which we here define as "long distal breaks" (LDBs), were further observed in GI/(I)GI (Fig. 2e). Type II deletions compared to type I deletions (not involving *RB1*) were significantly enriched in GI/(I)GI (55%) vs. (I)EMT-L/EMT-L (37.3%) ($p = 0.02$, Fisher's exact test (two-sided)). LDBs involving or exceeding the majority of cytoband 13q21.1 (distal of 54.7 mb) were significantly more frequent in GI/(I)GI (17.6%) vs. (I)EMT-L/EMT-L (7.1%) (p = 0.03, Fisher's exact test (two-sided)). LDBs and type II deletions showed a significant co-occurrence (87.5%) compared to LDBs and type I deletions (12.5%) ($p < 0.001$, one-sample proportions test with continuity correction) (Fig. 2e).

Taken together, the GI/(I)GI CLL subtype shows increased genomic complexity and selection of distinct chromosomal aberrations which may contribute to the disruption of genome integrity.

**CLL cases with genomic instability show alterations in processes protecting genome integrity.** Research into genomic instability in CLL has been primarily focused on functional loss of

*TP53* or *ATM* and the consequent dysfunctional DDR. Conversely, here we observed enrichment of DNA repair signatures in GI/(I)GI (Figs. 2a and 3a–d) and upregulation of *ATM* and *TP53* (Fig. 2c), indicating increased DDR activation. Importantly, upregulation of p53 and phospho-p53 protein levels was confirmed in genomically instable cases without recurrent gene mutations or chromosomal aberrations other than del(13q) (Fig. 3e, Supplementary Fig. 3m), confirming a continuous activation independent of such lesions.

We identified numerous interdependent alterations increasing genomic instability in GI/(I)GI (summarized in Fig. 3f). Critical pathogenic events involved telomere erosion (Fig. 1c) and alterations of the shelterin complex. *POT1* mutations (Supplementary Fig. 2c) are associated with telomeric abnormalities, chromosome breaks or fusions[20], and *POT1* overexpression (Supplementary Fig. 2d) implicated an increased requirement for telomere protection. In addition, *BUB1B* and *KNSTRN*, involved in correct chromosome segregation[21,22], were identified by GISTIC as putative gene targets for del(15q15.1) (Fig. 2b). We further found alterations of genes involved in regulating cell cycle checkpoints (Fig. 2d, e, Supplementary Fig. 3a, d, m) and numerous alterations impairing

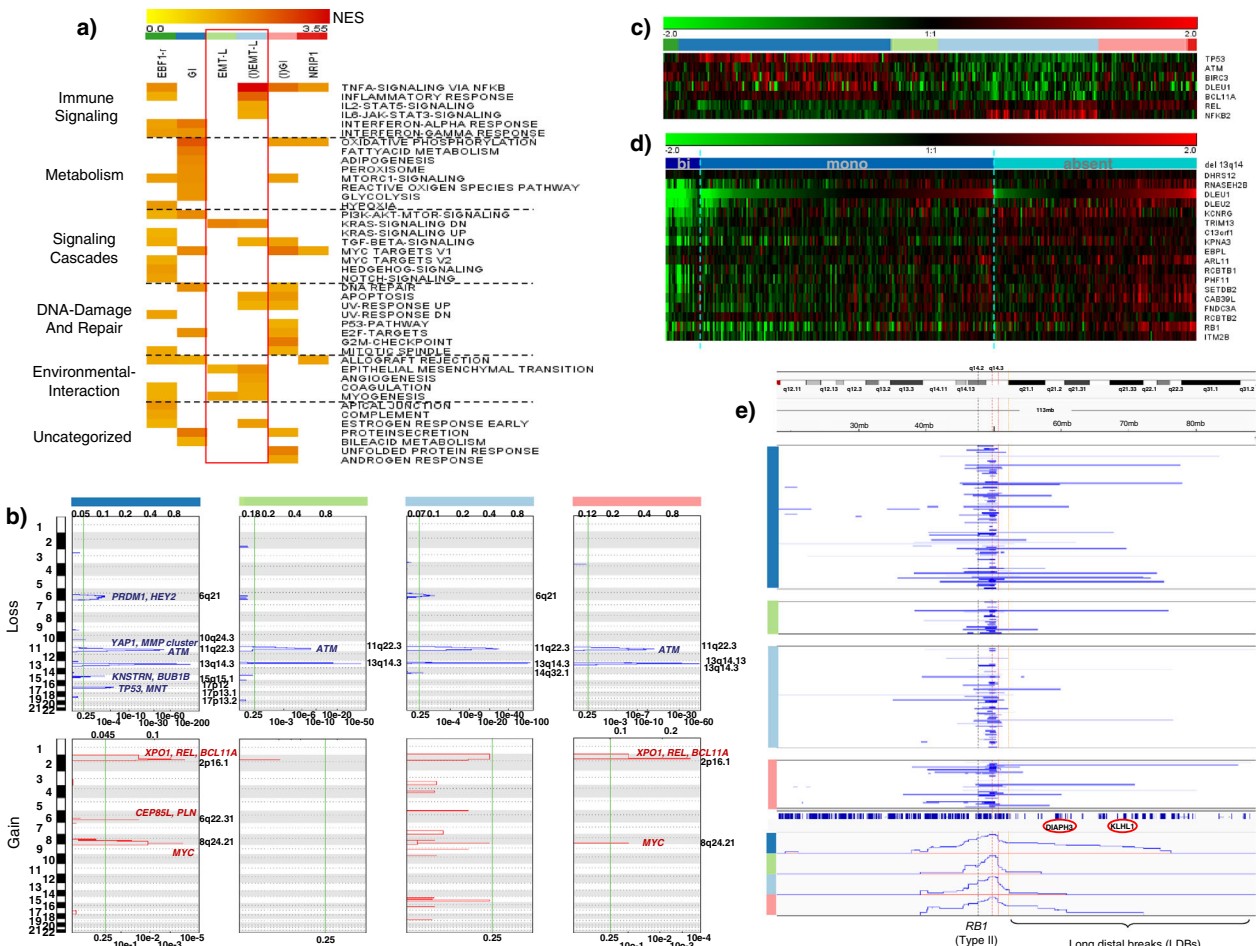

**Fig. 2 Pathway activation and genetic alterations in CLL subtypes. a** Heatmap showing overrepresented gene sets (FDR < 0.05) identified through GSEA. The intensity range of normalized enrichment scores (NES) illustrates the degree of enrichment in CLL subtypes. Gene sets are grouped together according to the biological context. CLL subtype color code defined in Fig. 2a applies for Fig. 2a–c, e. **b** GISTIC analysis of copy number alterations (CNA). Chromosomal positions (1–22) on the y-axis (left) indicate losses (blue, upper panels) or gains (red, lower panels) for major clusters. Affected genes representing CNA targets within biological networks (such as *YAP1*) are shown for respective peaks. The most significant chromosomal peaks for major clusters are indicated on the right of each panel. GISTIC q-values at each locus are plotted from left to right on a log scale (bottom of each panel). Altered regions with FDR q ≤ 0.25 (vertical green line) are considered significant. GISTIC G-Scores (amplitude of the aberration × frequency of its occurrence across samples) are plotted on top of the panels. **c** Heatmap showing GEP of genes located within or adjacent to the minimally deleted or minimally gained regions of recurrent aberrations (n = 337). FDRs for DEGs ((I)EMT-L vs. GI) are highly significant (q < 1e−07). **d** Heatmap showing GEP of genes located at/adjacent to the minimally deleted region on 13q (n = 335). The blue color code indicates deletions (dark blue: biallelic; light blue: monoallelic) and the absence of del(13q) (cyan blue). Genes are ordered corresponding to chromosomal positions for the region between *DLEU1* and *RB1*. **e** Visualization of del(13q) per case (blue horizontal lines). y-Axis: cluster color code, x-axis: representative genes and topography for the cumulative coverage of segment breaks per cluster. A vertical black dotted line indicates the *RB1* locus. Vertical red dotted lines indicate the majority of distal losses (around 50–51 mb). Losses extending to the distal end of cytoband 13q14.3 (orange dotted line) are variably distributed. LDBs involve/exceed the majority of cytoband 13q21.1 (distal of 54.7 mb). Biallelic deletions of 13q14 mostly cover a small region and rarely occur together with larger 13q deletions. For Fig. 2a–e, data within individual figures derives from biologically independent samples.

p53 and apoptosis (Figs. 2b, 3h, and 4d, Supplementary Fig. 3a, c, m). Enrichment of DEGs for specific chromosomal regions supported complex alterations of tumor suppressors (Supplementary Fig. 3e, f, g). Further, imbalanced MYC networks (Fig. 3g) involved increased activation (Figs. 2a, b, 3h, Supplementary Fig. 3b, d, m) or loss of repressors (Figs. 1b and 2b, Supplementary Fig. 2c, d) of MYC family members. In line with increased genomic instability, we observed a higher frequency and complexity of chromosomal aberrations in GI compared to (I)EMT-L cases after genotoxic stress (p = 0.03, Mann–Whitney (two-sided)) (Fig. 3i).

**Signatures of mutational processes highlight the pathogenic role of DNA repair deficiency and activation-induced cytidine deaminase in genomically unstable CLL.** To better characterize pathogenic processes in CLL subtypes, we analyzed CLL8 cases with existing WES data (n = 171, CD19 sorted) for mutational processes in cancer[23], referenced in the COSMIC database. Signature projections revealed strong activation of signature 6 (defective mismatch repair (MMR), microsatellite unstable tumors), along with other mutational processes, including signature 3 (defective double-strand break (DSB)-repair) and 15 (defective MMR, observed in stomach and lung cancer), in GI (Fig. 3j). Activation of signature 2, attributed

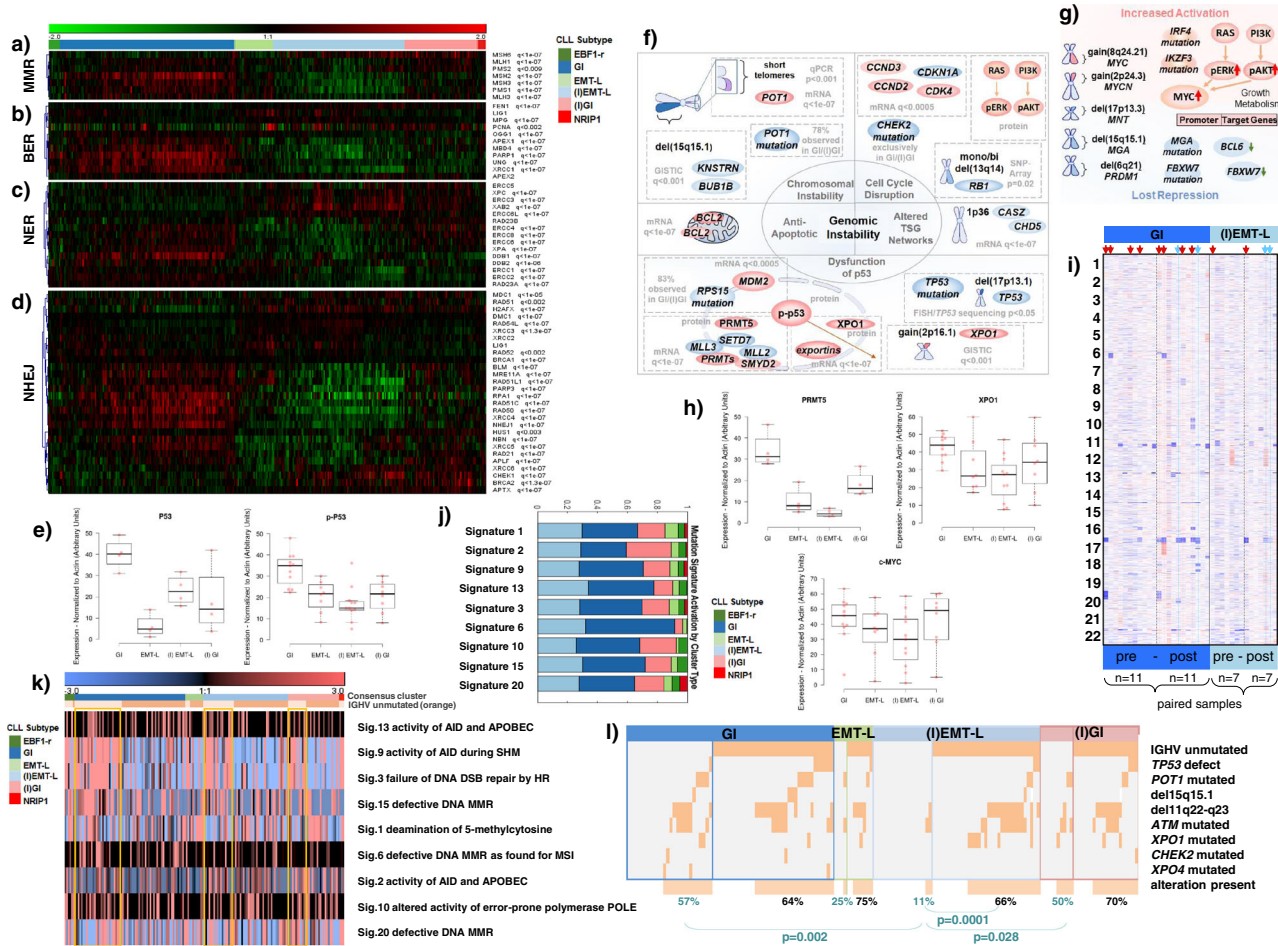

**Fig. 3 Biological processes operative in genomically instable CLL. a–d** Heatmap showing expression profiles ($n = 337$) for genes involved in **a** mismatch repair (MMR), **b** base excision repair (BER), **c** nucleotide excision repair (NER), **d** non-homologous end joining (NHEJ). FDRs of DEGs (GI vs. (I)EMT-L) are indicated (q). Single genes may be involved in multiple processes. **e** Protein expression in CLL subtypes; p53 ($n = 4$ each), phospho-p53 (GI/(I)EMT-L: $n = 11$ each, (I)GI/EMT-L: $n = 8$ each) (normalized to actin). **f, g** Models summarizing alterations which contribute to **f** genomic instability, **g** activation of MYC family members in GI/(I)GI. Source/method (e.g., mRNA/GISTIC) and significance/frequencies are shown in gray, along with estimated mode of regulation/biological effect (red: increase/activation; blue: decrease/inactivation). **h** Protein expression in CLL subtypes; PRMT5 ($n = 4$ each), XPO1/ cMYC (GI/(I)EMT-L: $n = 11$ each, (I)GI/EMT-L: $n = 8$ each) (normalized to actin). **i** Heatmap showing CNAs (paired CLL8 cases; before treatment (pre), at relapse (post)). CNAs are frequent at relapse (TP53 wild-type: GI/(I)EMT-L; $p < 0.05$, Wilcoxon signed-rank test (two-sided)). GI cases show more aberrations (mean) before ((I)EMT-L: 0.83; GI: 1.71) and after treatment ((I)EMT-L: 2.17; GI: 3.43), with considerable increase after treatment when TP53 mutations are included (GI(pre): 1.91, GI(post): 4.36; $p < 0.01$, Wilcoxon signed-rank test (two-sided)). Arrowheads highlight TP53 inactivation (preexisting: red; acquired: blue). GI alterations often involve chromosomes other than 13, 12, 11, and 17. **j** Fraction of signature activations in CLL subtypes (EBF1-r $n = 6$, GI $n = 68$, EMT-L $n = 11$, (I)EMT-L $n = 52$, (I)GI $n = 31$, NRIP1 $n = 3$) and **k** activation levels of mutational signatures (median centered) with order/color code according to clusters and IGHV status (light orange: IGHV mutated, yellow box: GI/(I)GI/(I)EMT-L. IGHV mutated cases show higher activations for signatures 9, 3, 15, and 20. **l** Alterations in DNA-damage response genes are more frequent in IGHV mutated GI/(I)GI cases (57%/ 50% ≥ one alteration) vs. IGHV mutated (I)EMT-L cases (11% ≥ one alteration) ($p < 0.05$, Mann–Whitney (two-sided)). IGHV unmutated cases (either subtype) have similar frequencies (64–70% ≥ one alteration). Only cases with WES for respective genes and known TP53 status (excluding EBF1-r/NRIP1) were used ($n = 156$). For Fig. 3a–l, data within individual figures derives from biologically independent samples. For boxplots, centerline, box limits, and whiskers represent the median, 25th, and 75th percentiles and 1.5× interquartile range, respectively.

to the activity of APOBEC family members, was increased in (I)GI compared to GI ($p = 0.01$, Mann–Whitney (two-sided)) (Fig. 3j, Supplementary Fig. 3h), irrespective of APOBEC expression (Supplementary Fig. 3l). This suggests that pathogenic deamination processes inducing DNA lesions are heterogeneous in genomically instable CLL. Signature 9 is attributed to the involvement of polymerase η in processing activation-induced cytidine deaminase (AID, encoded by AICDA) mediated cytidine deamination[24,25] during somatic hypermutation (SHM) and was specific for IGHV mutated cases ($p = 2.5e{-}15$, Mann–Whitney (two-sided)) (Fig. 3k, Supplementary Fig. 3i). Signature 9 was higher in GI compared to (I)EMT-L ($p = 0.03$, Mann–Whitney (two-sided)) (Supplementary Fig. 3j),

while the distribution of IGHV mutated cases was balanced across subtypes (GI: 43%, (I)EMT-L: 37%, (I)GI: 36%). Amplifications of MYC (8q24.21) were most frequently observed in GI/(I)GI cases, in line with the role of AID and possibly other APOBEC family members in mediating amplifications and translocations of this region. Moreover, IGHV mutated cases showed significantly higher activation of signatures 15 ($p = 0.01$), 3 and 20 ($p < 0.005$) (Mann–Whitney (two-sided)), indicating defective DNA repair (Fig. 3k). Based on this observation, we considered if the error rate through defective DNA repair in GI might be exacerbated in situations of increased AID activity (and the respective AID-induced mismatches or DSB in non-Ig loci). We observed a trend for

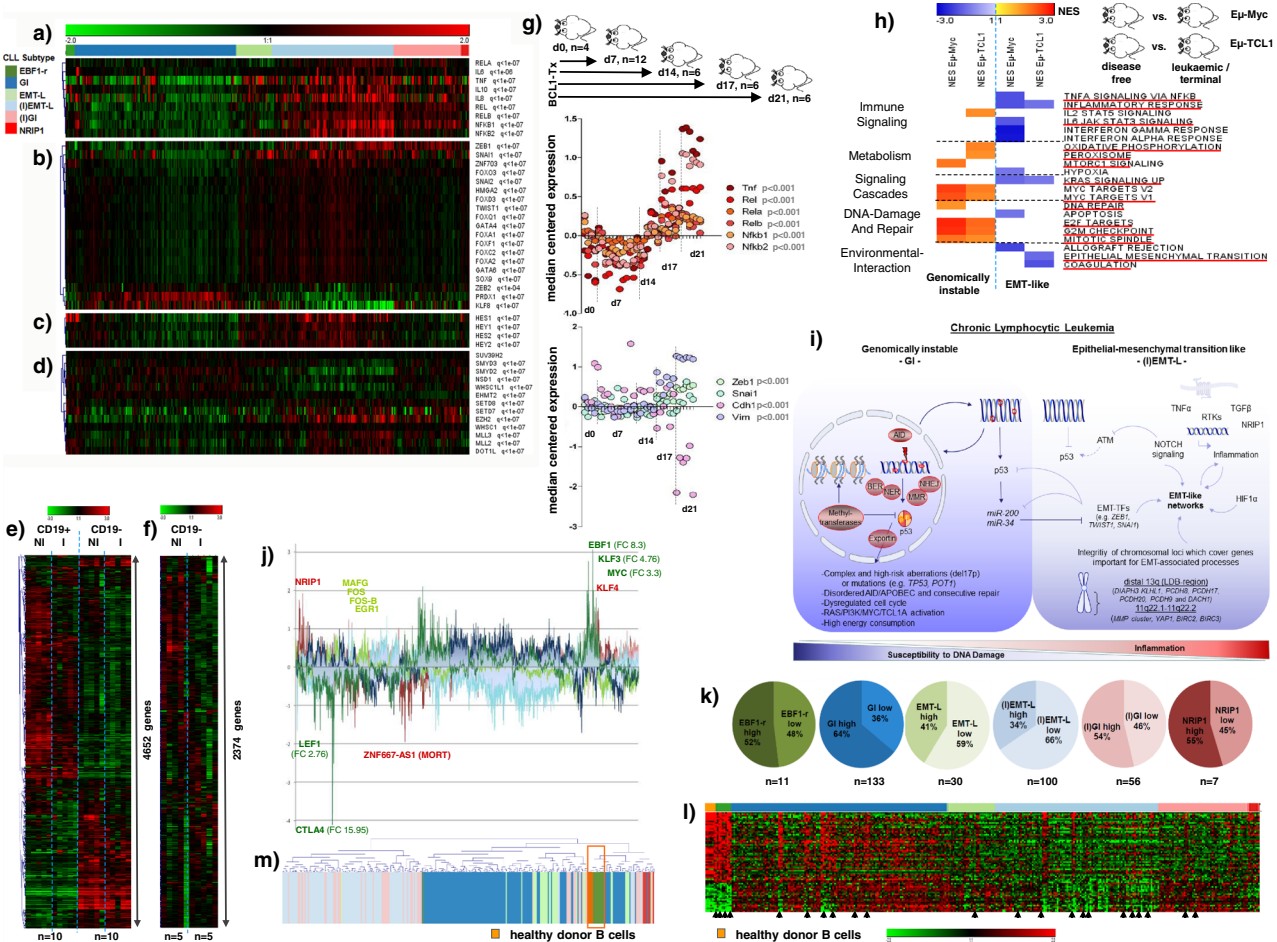

**Fig. 4 Biological processes operative in CLL with EMT-like networks. a–d** Heatmap showing expression profiles (n = 337) for **a** genes indicating activated TNFα/NF-kB signaling, **b** EMT-TFs, **c** NOTCH target genes (intensity range −1:1), **d** histone lysine methyltransferases. FDRs of DEGs (GI vs. (I)EMT-L) are indicated on the right (q). CLL subtype color code defined in Fig. 4a applies for Fig. 4a–d, j–m. **e** GEP of the CD19 positive (+) and negative (−) compartment from CLL samples with inflammatory (I) and non-inflammatory (NI) signatures. **f** GEP of 2374 variably expressed genes for the CD19 negative fraction. **g** Tumor GEP indicating activated TNFα/NF-kB signaling and induction of EMT-like programs after BCL₁ tumor transplantation. y-axis: median centered expression, x-axis: days (d) after transplantation of individual samples (p < 0.05 shown for d7 vs. d21, Mann–Whitney (two-sided)). **h** Heatmap showing gene set enrichment, characteristic for CLL with genomic instability or activation of EMT-like programs, in Eμ-Myc and Eμ-TCL1 mice. Gene sets were identified through GSEA (FDR < 0.05) on proteome profiles of splenic tumor cells from leukemic or terminal Eμ-Myc and Eμ-TCL1 mice (n = 4, in two pools) compared with B cells of tumor-free wild-type mice (n = 12, in two pools). Color-coded normalized enrichment scores (NES) (degree of enrichment; positive: yellow to red; negative: blue, light to dark). Gene sets are grouped according to the biological context. **i** Model illustrating biologic characteristics and regulatory interplay of processes in major subgroups (GI and (I) EMT-L), as specified in respective results sections. **j** Expression profiles for 2359 variably expressed genes (SD > 0.5). Genes with the highest significance (q < 1e −05) for the EMT-L, EBF1-r, and NRIP1 cluster are indicated. Fold change (FC) is indicated for EBF1-r specific genes (EBF1-r vs. all other). **k** Piecharts illustrating global gene expression, percentages indicating over- or under-expression in relation to the median expression per gene across the dataset. **l** Heatmap showing genes (n = 69) with strongest differential expression (q ≤ 0.05, FC ≥ 2) between the EBF1-r vs. all other clusters. CD19 sorted healthy donor B cells are included (orange). Arrowheads indicate cases with tri(12). **m** Agglomerative hierarchical clustering (2359 genes, Pearson complete) for n = 337 CLL and n = 5 healthy donor B cells (orange). For Fig. 4a–f, h–m, data within individual figures derives from biologically independent samples.

higher activation of DNA damage associated signatures 3 (defective DSB-repair) and 15 (defective MMR) and significantly higher activation of signature 6 (defective MMR) (p = 0.02, Mann–Whitney (two-sided)) in IGHV mutated GI cases compared to IGHV mutated (I)EMT-L cases, while activations in IGHV unmutated cases of either subtype were low (Fig. 3k, Supplementary Fig. 3k). Genomic alterations implicated in genomic instability had a significantly lower incidence in IGHV mutated (I)EMT-L cases compared to IGHV mutated GI/(I)GI cases (Fig. 3l), which supports a selective vulnerability in the context of AID/APOBEC activation and insufficient MMR. Conversely, when considering IGHV unmutated cases, which show a high frequency of alterations in genes like TP53, ATM, or POT1 (Fig. 3l), mutational signatures associated with AID and MMR

deficiency were generally low (Fig. 3k). We subsequently assessed the impact of AID-mediated induction of genomic instability in vitro which validated such deleterious effects on the genome (Supplementary Fig. 4a–i).

Together, these data show that distinct subgroups of CLL exhibit an increased accumulation of genomic lesions in association with mutational processes indicating deficient DNA repair and increased AID activity. Our observations indicate that the identified mutational processes represent independent pathogenic mechanisms in GI/(I)GI subtypes.

**EMT-like differentiation evolves in conjunction with inflammation and genomic stability.** As described above, we observed

enrichment of gene sets characteristic for EMT in cases that showed higher genomic stability (Fig. 2a). During EMT, cells lose adhesion and gain invasive properties to exit from the surrounding tissue, as found for metastasis. Alterations in the EMT-like subgroup reflected central hallmarks of EMT such as extracellular matrix remodeling (Supplementary Fig. 5a) and increased cell motility (Supplementary Fig. 5b).

Transcriptional signatures indicating immune signaling and TNFα-mediated inflammation were specifically upregulated in cases with EMT-like networks (Figs. 2a and 4a, Supplementary Fig. 5c) and correlated with upregulation of *NRIP1*, a coactivator of NF-kB-mediated inflammation (Supplementary Fig. 2a). Besides inflammation, which serves as a strong EMT inducer, we were able to validate other EMT-inducing alterations like increased TGF-β signaling (Fig. 2a) and *HIF1α* upregulation (Supplementary Fig. 5e) for CLL cases with EMT-like networks.

Overexpression of EMT-associated transcription factors (EMT-TFs) (e.g., *ZEB1*, *SNAI1*, and *TWIST1*) (Fig. 4b) and receptor tyrosine kinases (Supplementary Fig. 5d) were further confirmative for induction of EMT-like cellular programs. Notably, EMT-TFs showed a similar expression in different compartments like lymph nodes, bone marrow, and peripheral blood (Supplementary Fig. 5c). We also found overexpression of *EZH2*[26] and *SETD7*[27], which may enhance NF-kB- (Fig. 4a) and NOTCH-(Fig. 4c) mediated EMT, along with other lysine methyltransferases (Fig. 4d). Inflammatory features were associated with lower peripheral WBC counts (Fig. 1d) supporting distinct migratory properties and environmental interaction. Notably, GEP of the CD19 negative cellular fraction (comprising the nonmalignant blood component, i.e., monocytes, T and NK cells) showed unique signatures which reliably discriminated between patients belonging to the inflammatory/EMT-L or non-inflammatory/GI subtype (Fig. 4e, f). These findings, therefore, offer evidence for a subtype-specific, CLL-mediated impact on nonmalignant immune cells and altered environmental interaction.

We further identified several genes in recurrently deleted regions, including *YAP1* and an *MMP* cluster on 11q22.1–q22.2 (Supplementary Fig. 2e–h) or genes residing in LDB-regions on 13q, like protocadherins (Fig. 2e), which are closely linked with the EMT process. Since the respective alterations were primarily observed in genomically instable CLL, the integrity of these regions seems indispensable for the differentiation towards EMT-like networks. In conclusion, CLL with EMT-like changes constitutes a distinct biologic subgroup with a differentiated impact on the environment.

**EMT-like differentiation can be induced in lymphoma and shows reciprocal inhibition with genomic instability**. Since CLL cases with EMT-like networks show strong transcriptional signatures indicating inflammation and immunological response, we next aimed to validate the effects of inflammation on EMT-induction in vivo by utilizing a syngeneic BCL$_1$ lymphoma transplant mouse model. The BCL$_1$ tumor is a syngeneic lymphoma of BALB/c origin and transplantation results in a typical B cell leukemia/lymphoma characterized by splenomegaly, peripheral blood lymphocytosis, and death of all tumor-bearing mice. This model was used as it reflects major hallmarks of the (I)EMT-L subtype: (1) lymphoma cells experience strong environmental stimulus during tumor development and migration to lymphoid organs and, (2) tumor transplantation itself is associated with a heavy inflammatory response. Notably, spleen tumor samples obtained at defined time points after transplantation showed dynamic GEP changes confirming inflammation and the associated induction of EMT-like networks as indicated by

upregulation of *Vim*, the EMT-TFs *Zeb1*, *Snai1*, and downregulation of *Cdh1*, all characteristic of EMT (Fig. 4g).

Regarding the rare occurrence of alterations associated with genomic instability in cases with EMT-like networks, we next hypothesized that p53 activation and an increased DDR may inhibit induction of EMT-like programs. Specifically, (I)EMT-L cases exhibited low p53 and phospho-p53 levels (Fig. 3e, Supplementary Fig. 3m), and related pathway activation (Fig. 2a, c) alongside inverse correlation of *TP53* and *ZEB1* expression (Supplementary Fig. 5f). Lymphoma cells showed *miR-200c* induction (Supplementary Fig. 5g) while the respective targets, EMT-TFs *ZEB1* and *TWIST1*, decreased after ionizing radiation (Supplementary Fig. 5h), in line with the p53-*miR-200c*-*ZEB1* mediated suppression of EMT in solid tumors[28]. Further, increased NOTCH signaling (Fig. 4c) may stabilize EMT-like networks through DDR suppression (Fig. 3a–e), as previously reported[29].

We subsequently aimed to validate the inhibitory effects of genomic instability on EMT-like differentiation in vivo. Among the different genes found to be involved in genomically instable cases, *TCL1A* and *MYC* were amplified and/or consistently upregulated in the GI/(I)GI subgroup and the corresponding pathway has been identified as a central element for this pathogenic network (Fig. 3h, Supplementary Fig. 6b). Both genes represent single oncogenic drivers and their overexpression in murine models leads to aggressive lymphomas with rapid proliferation and genomic instability.

With that in mind, we used the two corresponding models: (1) Eμ-Myc[30], where the c-Myc oncogene is placed under the control of the immunoglobulin enhancer to induce a highly aggressive B-lymphoid malignancy, and (2) Eμ-TCL1[31], where the TCL1 oncogene is driven by the immunoglobulin enhancer and represents a well-established model of CLL. Tumors derived from these models were specifically assessed for c-Myc- and TCL1-induced pathway alterations by proteome profiling. Confirming the observations made in the human samples, we identified proteome profiles reflecting network activation observed for genomically instable cases while processes characteristic for the EMT-like subgroup were downregulated (Fig. 4h). Alterations leading to genomic instability, therefore, contribute to the inhibition of EMT-like networks.

EMT-transition in vivo was recently found to occur through a multistep process rather than a binary switch[32]. We assessed if such plasticity was present and if the transition to EMT-like networks could be modulated, in aggressive Eμ-TCL1 tumors, similar to observations made for the BCL$_1$ model. Generating strong EMT-inducing stimuli with repetitive cycles of inflammation and environmental interaction through serial transplantations (Supplementary Fig. 5i), we were able to mimic EMT plasticity in Eμ-TCL1 tumor cells (Supplementary Fig. 5j, k).

These findings provide evidence that induction and maintenance of EMT-like networks require convergence of strong EMT-inducing stimuli, while alterations associated with genomic instability and increasing aggressiveness inhibit EMT-like differentiation as depicted in our model (Fig. 4i).

**Pathogenic networks in CLL are not epigenetically controlled through DNA methylation**. Chromatin organization influences transcriptional activity during differentiated biologic processes, and altered states can initiate or maintain pathogenic conditions such as EMT and genomic instability[33–36]. We observed characteristic profiles suggesting a heterogeneous transcriptional activity in CLL subtypes (Fig. 4j, k) supported through distinct patterns of epigenetic modifiers. These included *MAFG/DNMT3B*[37] (Supplementary Fig. 6a, c), *TCL1A/DNMT3A*[38] (Supplementary Fig. 6b, c), TGF-β

signaling[39] (Fig. 2a), histone deacetylases (Supplementary Fig. 6d), chromodomain–helicase–DNA-binding proteins (Supplementary Fig. 6e) and others (Supplementary Fig. 6f). We also observed a stringent association of these genes and the cluster hierarchy, irrespective of chromosomal alterations, particularly exemplified for tri(12) cases. Tri(12) cases exhibited highly distinct transcriptional profiles forming a single cluster (EBF1-r) or leading to clustered enrichment in GI and (I)EMT-L (Figs. 1b and 4l, Supplementary Fig. 6g, h). *EBF1* showed the strongest overexpression in EBF1-r (Fig. 4j) and in agreement with its crucial involvement in the differentiation of B cells[40], EBF1-r cases stringently clustered with healthy donor B cells (Fig. 4m).

Notwithstanding such extensive transcriptional reprogramming toward mature B cells, tri(12) cases retained distinct profiles of epigenetic modifiers reflecting the respective cluster hierarchy (Supplementary Fig. 6i). To investigate if epigenetic modification through DNA methylation may contribute to pathogenic network profiles, we analyzed reduced representation bisulfite sequencing (RRBS) data for $n = 182$ matched cases. Robust methylation differences across groups were not observed (Supplementary Fig. 6j) for RRBS data. While processes involving AID/APOBEC family members and BER may actively promote demethylation in genomically instable cases, we observed significantly higher expression levels of DNA-demethylases in (I)EMT-L compared to GI (Supplementary Fig. 6k).

### Genomic instability is associated with poor prognosis in CLL.
To define the clinical impact resulting from the underlying biology of identified CLL subtypes, we assessed the clinical course in previously untreated patients of the CLL8 trial. Complete and partial remissions were similar for identified subtypes (Supplementary Tables 4 and 5). Progression-free survival (PFS) was shortest for chemotherapy treatment in genomically instable cases but increased considerably when fludarabine and cyclophosphamide (FC) was combined with rituximab (R), with the strongest overall benefit observed for (I)GI (GI: median PFS 27.8 months (FC) vs. 42.4 months (FCR), HR: 0.55 (95% CI: 0.37–0.82), $p = 0.004$; (I)GI: median PFS 22.8 months (FC) vs. 68.1 months (FCR), HR: 0.30 (95% CI: 0.15–0.60), $p = 0.001$) (Fig. 5a, Supplementary Fig. 7a). In comparison, PFS was considerably longer for FC treatment in (I)EMT-L/EMT-L cases, but lacked a similar increase in efficacy when rituximab was added ((I)EMT-L: median PFS 36.1 months (FC) vs. 52 months (FCR), HR: 0.86 (95% CI: 0.53–1.41), $p = 0.56$; EMT-L: median PFS 45.5 months (FC) vs. 65.2 months (FCR), HR: 0.63 (95% CI: 0.25–1.56), $p = 0.31$) (Fig. 5a, Supplementary Fig. 7a).

Notably, overall survival (OS) was significantly improved in (I)GI when FC was combined with rituximab (median OS not reached (FCR) vs. 56.6 months (FC), HR: 0.32 (95% CI: 0.13–0.79), $p = 0.013$) (Fig. 5a, Supplementary Fig. 7b).

To further elucidate the clinical impact of the underlying biology, we performed survival analyses for genetically defined categories. We first assessed the clinical outcome in subtypes with regard to the IGHV mutation status. As previously described, identified patterns for mutational signatures support a selective vulnerability in the context of AID activation and insufficient MMR in GI/(I)GI cases (Fig. 3k, l). Confirmatory, IGHV mutated (I)EMT-L cases showed the longest PFS and OS rates (Fig. 5b, Supplementary Fig. 8). Survival differences were especially pronounced for FC treatment, showing a considerably shorter median PFS for IGHV mutated GI cases compared to IGHV mutated (I)EMT-L cases (29.1 months vs. not reached, HR: 0.29 (95% CI: 0.11–0.77), $p = 0.013$), while median PFS in IGHV mutated GI cases was similar to IGHV unmutated cases of both the GI and (I)EMT-L subtype (24.4 and 27.8 months) (Supplementary Fig. 8c, Supplementary Table 6).

Cases with *TP53* alterations and wild-type *TP53* show a high transcriptional homogeneity in the GI/(I)GI cluster (Fig. 1b), reflecting similar biology. We next confirmed that GI compared to (I)EMT-L cases also show a poorer clinical course when segregated for chromosomal aberrations or *TP53* wild-type status (Supplementary Fig. 9, 10, Supplementary Table 7). Characteristics, other than *TP53* defect, were homogenously distributed between both groups (Supplementary Table 2). The prognostic impact of biological subtypes was further validated in an extended analysis for molecularly defined subcategories. While alterations of *TP53* and *ATM* were found to associate with a poor clinical course, independent of the respective biological subtype, we observed a strong impact on outcome in *TP53* and *ATM* wild-type cases and with regard to the *SF3B1* mutation status (Fig. 5c, d). The addition of rituximab considerably improved outcome in GI, whereas (I)EMT-L cases in contrast consistently lacked a similar increase of efficacy (Fig. 5a, c, d, Supplementary Figs. 7–12). In line with imbalanced rates of lethal sepsis (Supplementary Table 8), representing the fulminant release of cytokines from the immune system, and distinct expression profiles for CD19- non-malignant immune cells (Fig. 4f), differential treatment efficacy for the addition of rituximab strongly supports a heterogeneous responsiveness of the immune system in identified biological subgroups.

### Validation of CLL subtypes and prognostic impact in the REACH study cohort.
We next validated the major subtypes in an independent phase III trial cohort of relapsed CLL patients (REACH, $n = 300$) and a second internal validation set of previously unexamined, U-CLL8 samples ($n = 89$) (Fig. 6a, c, Supplementary Fig. 13a–c, Supplementary Table 9).

REACH was analyzed complementary to CLL8 by using CC on variably expressed genes with SD > 0.5 (Fig. 6a), which reliably identified the same biological categories (Fig. 6b, c, Supplementary Fig. 13a). Subtype-specific expression patterns were also found when hierarchical clustering was applied on the internal U-CLL8 validation set (Supplementary Fig. 13b). Specific analysis of expression profiles of genes associated with increased DDR, alternative mechanisms for p53 inactivation, and distribution of cases with *TP53* defect further validated the CLL subtypes in the REACH cohort (Supplementary Fig. 13c, Supplementary Table 9).

Increased transcriptional homogeneity and co-clustering of cases classified as "GI/(I)GI" supported the selection of unifying features after treatment (Fig. 6a, c, Supplementary Fig. 13c–e). The prognostic impact remained identical as observed for (I)EMT-L and GI in CLL8 (Supplementary Fig. 14a, b). As hypothesized from the biological context, median PFS rates in "GI" cases without *TP53* defect resembled those of cases with *TP53* defect (19.9 vs. 17.1 months), in contrast to "(I)EMT-L" cases (median PFS 38 months, $p < 0.0001$) (Fig. 6d). Median OS was shortest for cases with *TP53* defect (35 months), followed by "GI" (68 months, $p < 0.0001$) and "(I)EMT-L" (not reached) (Fig. 6e).

Notably, the GI subtype was identified in a multivariate Cox proportional hazards regression model, along with unmutated IGHV and del(17p), as an independent adverse prognostic factor associated with short PFS in relapsed CLL cases (REACH) (Supplementary Table 10).

### Discussion
In this study we identified genetically and clinically distinct CLL subgroups, comprising genomic instability or activation of EMT-like networks, extending the current perspective on disease pathogenesis, progression, and resistance.

While we observed higher leukocyte counts for the CLL8 discovery cohort of CD19 sorted CLL cases, likely through the

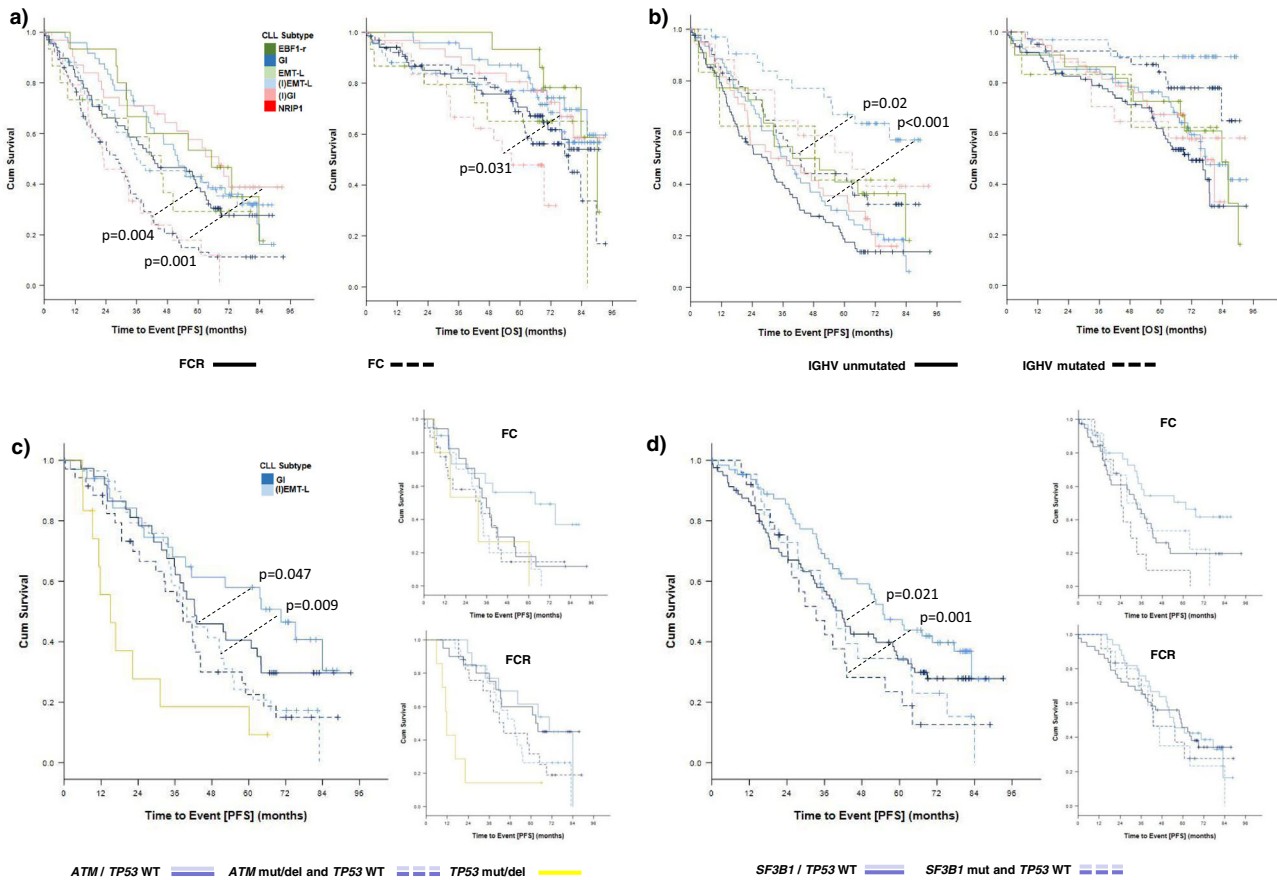

**Fig. 5 CLL subtype, genetic markers, and treatment outcome in CLL8. a** PFS (left) and OS (right) according to treatment arm (FC: dotted line; FCR: continuous line) and subtype (color-coded) (*n* = 319). CLL subtype color code defined in Fig. 5a applies for Fig. 5a, b. **b** PFS (left) and OS (right) according to the IGHV mutation status and subtype (color-coded) (both treatment arms) (*n* = 310). **c** PFS according to subtypes; GI (dark blue) and (I)EMT-L (light blue), *TP53*, and *ATM* mutation and/or deletion status (shown for all cases and individual treatment arms) (*n* = 147). CLL subtype color code defined in Fig. 5c applies for Fig. 5c, d. **d** PFS in *TP53* wild-type cases according to *SF3B1* mutation status for GI (dark blue) or (I)EMT-L (light blue) (*n* = 193). For Fig. 5a–d, the log-rank test was used to compare the survival distributions. Data within individual figures derives from biologically independent samples.

selection of samples with abundant material for multiple analyses, patient characteristics and especially high-risk markers showed a well-balanced distribution representative of the full trial population.

As implied from the identified mutational signatures, genomic instability may be present before malignant transformation or in the early phase of the disease and facilitated by defective DNA repair mechanism and activation of AID during SHM[41]. Furthermore, AID may be reactivated during the disease course[42,43] and add to the acquisition of genomic lesions and clonal evolution in the GI subtype.

The specific association with the GI/(I)GI subtype further highlights the role of arginine and lysine methyltransferases in genomically instable CLL previously found to promote lymphomagenesis or induce genomic instability in cancer[44–48].

While we could not detect differences for promoter or gene body methylation in identified subgroups, gene-specific regulation of methylation or demethylation dynamics in distinct regions may still impact pathogenic networks. Our data further support a broad involvement of other epigenetic modifiers on subtype-specific chromatin organization, which may impact genomic stability similar to DNA methylation[33–35].

Development of treatment-resistant CLL has been associated with the inactivation of *TP53*, *ATM*, and correspondingly a deficient DDR[3–6]. However, here we show rather that genomically instable CLL exhibit activation but insufficient execution of DNA-repair programs, irrespective of *TP53* status. We have further classified multiple alterations contributing to genomic instability into distinct but interdependent processes. These involve disruption of telomere maintenance, DNA and chromosome integrity, altered DDR with insufficient DNA repair, MYC pathway activation, disrupted cell cycle checkpoints, and chromatin organization. Continued execution of these disordered processes, therefore, maintains genomic instability through the ongoing accumulation of genomic lesions.

Correspondingly, we show that therapy-associated genotoxic effects can aggravate genomic instability and worsen long-term treatment outcomes in such patients when receiving sequential therapies. In particular, the poor outcome in CLL with wild-type *TP53* and *ATM* highlights the importance of alternative mechanisms for the induction of genomic instability. We show that such cases benefit considerably from p53/DDR-independent treatments like rituximab, which mediates killing through non-DDR processes, dependent on Fc and FcγR engagement[49].

Contrasting the characteristics observed for GI, we identified (I)EMT-L cases to exhibit the most distant clustering and comprise highly differentiated biology from this subtype. While differentiation towards EMT is a well-known phenomenon in various cancers, EMT-like changes constitute a hitherto unappreciated aspect of CLL and were unexpected as metastasis and the underlying biologic processes seem redundant for leukemic dissemination. However, EMT-like features and direct

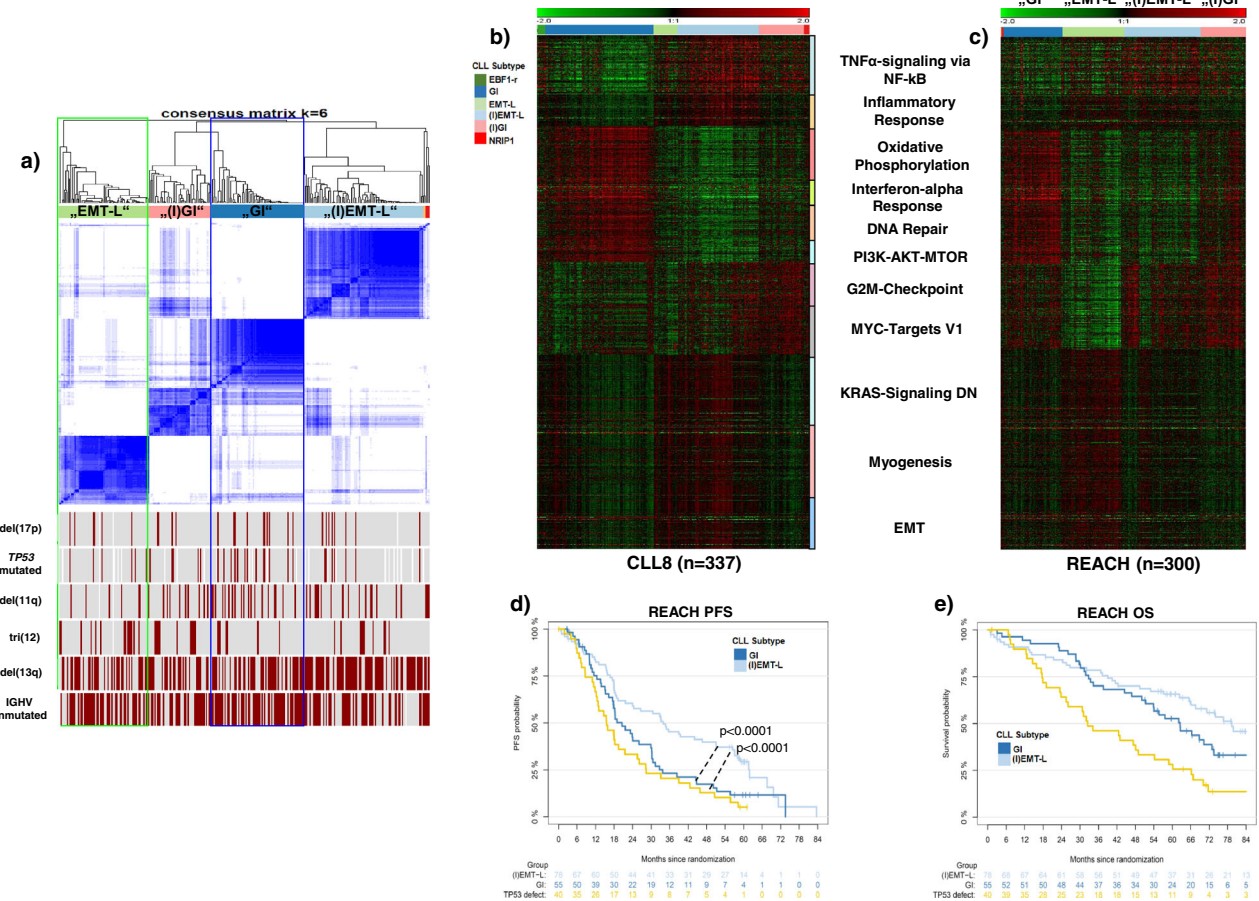

**Fig. 6 Validation of CLL subtypes and prognostic impact in the REACH trial cohort. a** Consensus heatmap showing the 4 major CLL subtypes identified in the REACH expression dataset ($n = 295$); only a few cases ($n = 5$) segregate in undefined clusters (red/orange). Recurrent alterations are depicted below, inactivation (del/mut) of *TP53* is heterogeneously distributed ("EMT-L": 11%, "(I)GI": 16%, "GI": 20%, "(I)EMT-L": 9%). **b** Heatmap depicting characteristic GEP in CLL8 (major core enrichment gene sets, 931 genes). **c** Heatmap showing expression of core enrichment gene sets (as used in (**b**)) for the REACH dataset. For better comparability, CLL8-complementary clusters (indicated by labels in quotation marks) are ordered in the same order as found by consensus clustering in CLL8. Increased biologic homogeneity is observed in "GI"/"(I)GI" cases and supported through co-clustering in (**a**). **d** PFS in the REACH dataset for all cases with *TP53* defect (yellow), "(I)EMT-L" and "GI" cases without *TP53* defect ($n = 173$). The significance level for GI vs. (I)EMT-L cases is calculated based on clustering from (**a**). **e** OS in the REACH dataset for all cases with *TP53* defect (yellow), "(I)EMT-L" and "GI" cases without *TP53* defect ($n = 173$). For **d**, **e**, the log-rank test was used to compare the survival distributions. Data within individual figures derives from biologically independent samples.

involvement of EMT-TFs were previously reported for pathogenic processes in hematologic malignancies. These involve regulation of migratory properties in myeloma[50,51], proliferative capacities and response to treatment in mantle cell lymphoma[52] and ALL[53], and regulation of immune checkpoints and aggressiveness in DLBCL[54,55]. Higher methylation levels were also found for the EMT-TF *TWIST2* in CLL with mutated IGHV[56] and in comparison to healthy donor cells[57]. Furthermore, EMT-TFs fulfill crucial roles in normal hematopoiesis and B cell maturation[58].

Since we identify interdependent changes in malignant B cells which resemble the EMT process we have called this subgroup and the corresponding specific alterations "EMT-like". Our observations do not put lineage-specific and phenotypic changes recognized in epithelial or mesenchymal tissues during EMT into focus but categorize cellular properties and processes (e.g., inflammatory changes, NOTCH signaling, and genomic stability) which are found likewise in either tissue context.

The EMT-like subgroup reflects various characteristics of CLL with increased environmental interaction via receptor or cytokine signaling and migratory properties[59–62]. EMT-TFs in CLL,

therefore, may regulate migratory capabilities to infiltrate lymphatic tissues or other pro-survival niches, similar to other cancers[32,63,64]. Multiple studies have shown the involvement of cytokines from the microenvironment to regulate tumor inflammation and induction of EMT. Conversely, EMT-TFs themselves can induce inflammation in cancer cells and shape the microenvironment accordingly[65]. EMT-like transcriptional programs in CLL, therefore, may be activated through inflammatory, HIF1α, NOTCH1, and other signaling cascades representing a convergence from multiple pathways during lymphomagenesis. Stable integration of these signals into activated EMT-like networks with a pro-survival advantage may consolidate such differentiation.

We observed a reduced benefit for rituximab in (I)EMT-L cases occurring in conjunction with a general NOTCH pathway activation but independent of NOTCH1 mutations previously associated with rituximab resistance[3]. Systemic effects on nontumor cells occurring in (I)EMT-L CLL may further elicit functional disruption of effector cells like macrophages, which execute treatment effects of rituximab[66]. In line with our findings, the presence of EMT-like GEPs has previously been linked with

immunosuppression in solid tumor studies using checkpoint inhibitors[67,68].

We provide experimental evidence from in vivo and in vitro studies that tumor aggressiveness and the extent of DDR directly regulate EMT-like networks through suppression of EMT-TFs. However, EMT-TFs can also act in the reverse direction to suppress the DDR. EMT-TFs were shown to downregulate p53 and diminish its transcriptional activity[69–71]. Further, expression of the *miR-200* and *miR-34* families is regulated by p53 and both target EMT-TFs[28,72–74], which themselves build tight regulatory loops with these miRs[75–77]. EMT-TFs protect against DNA damage in solid tumors, where *ZEB1* expression is inversely correlated with the incidence of CNAs and *TP53* mutations[78], which is again mirrored by our data. Such regulatory axes may strengthen the highly diversified biologic trajectories underlying the CLL subtypes. As recently reported, EMT-transition in solid tumors occurs through intermediate hybrid states, which exhibit distinct cellular properties and show a differentiated interaction with the microenvironment[32]. We show that EMT-like differentiation in lymphoma can be modulated, but comprises a distinct subgroup largely independent of sub-compartments like a lymph node, bone marrow, or peripheral blood.

In conclusion, this study extends the basis for understanding CLL pathogenesis and pathway dependencies that may be targeted by novel compounds. Identified molecular targets in a defined biologic context may further advance the development of new treatment strategies. Compound combinations targeting, for example, BCL2 and PRMT5 or XPO1, together with anti-CD20 monoclonal antibodies, may specifically synergize in genomically unstable cases. Future assessment of the subtype-related outcome in comprehensively characterized trial cohorts testing BCL2-, BTK- and other inhibitors in development will further elucidate the therapeutic potential of such treatment combinations.

## Methods

**Overview for conducted analyses and respective sample size.** Specimens were collected from patients registered on the CLL8[15] (first-line) and REACH[16] (relapse) trials after informed consent. All patients suffered from progressive disease with the need for treatment. Samples were CD19 sorted (CLL8 $n = 337$; REACH, $n = 300$) or unsorted (U-CLL8, $n = 89$). RNA and DNA were purified and assessed for integrity with routine protocols. Samples were used on the following experimental platforms: GEP (CLL8, $n = 426$; REACH, $n = 300$) on Human Exon 1.0 ST arrays (Affymetrix), SNP-array analysis (CLL8, $n = 309$ treatment-naive, of which $n = 18$ had a paired sample at relapse) on Human SNP Arrays 6.0 (Affymetrix), analysis of mutations and mutational processes (CLL8, $n = 171$) using WES (Illumina), RRBS for methylation analysis (CLL8, $n = 182$), western blots for validation were performed in selected cases without recurrent alterations. Fluorescence in situ hybridization (FISH), IGHV, and *TP53* mutation analyses were performed upon trial registration. The multiparameter analysis was conducted for CD19 sorted CLL8 samples and the distribution of genetic characteristics for analyzed cases was representative of the full CLL8 trial cohort (Supplementary Table 1). REACH and U-CLL8 samples were used as validation cohorts. The REACH cohort was chosen since it was designed complementary to CLL8 and ideally suited to independently validate identified transcriptional networks and related resistance mechanisms. U-CLL8 samples were chosen as internal validation set to assess the impact of lower tumor homogeneity and to exclude the possibility that transcriptional changes were induced through the process of CD19 sorting. Survival data were available with a median observation time of 5.9 (CLL8) and 4.9 (REACH) years. Pathogenic networks were validated using transgenic Eμ-Myc and Eμ-TCL1 mice or BCL$_1$ and Eμ-TCL1 tumor transplantation models. Details on methods, demographic and clinical characteristics are also provided in the CONSORT diagram (Fig. 7), sequential workflow and analysis is visualized in Figs. 1a, and 7 and Supplementary Fig. 1e.

**Patients and samples.** Analysis was conducted on peripheral blood samples from previously untreated CLL patients ($n = 426$) from the CLL8 trial, a prospective, international, multicenter, open-label, randomized phase 3 study, comparing first-line treatment with R-FC ($n = 408$) or FC ($n = 409$)[15]. Data on genomic aberrations and mutations, such as del(17p), del(11q), tri(12), del(13q), the IGHV, *TP53*, *SF3B1*, and *NOTCH1* mutational status, was included for analysis[3]. Patient characteristics of target analysis population of $n = 337$ CD19 sorted CLL used as discovery cohort is provided in Supplementary Table 1. Samples were collected at

enrollment on the CLL8 trial and selected for gene expression profiling based on availability and RNA quality.

For validation of discoveries, we further used an independent set of GEPs ($n = 300$) generated from CD19 sorted CLL patient samples of the REACH study, a prospective, international, multicenter, open-label, phase 3 study, in which patients with previously treated CLL were randomized to receive R-FC ($n = 276$) or FC alone ($n = 276$)[16]. Details on genetic characteristics and other variables for the CLL8 gene expression cohort are listed in Supplementary Table 1. The primary objective of the CLL8 and REACH study was to demonstrate superiority with regards to PFS for R-FC compared with FC alone. The study protocols were approved by institutional review boards at participating centers, and all patients gave written informed consent. Details on trial design and eligibility criteria and clinical outcome have been described elsewhere[15,16]. The studies were conducted in accordance with the Declaration of Helsinki. Details for each study are further provided online at the ClinicalTrials.gov (C T G) homepage. All baseline parameters including genetics, serum parameters (such as thymidine kinase, β2-microglobulin), and cell surface/membrane markers (such as ZAP-70) were performed in a centralized manner in accredited reference laboratories of the German CLL Study Group (GCLLSG) for the CLL8 trial, as outlined in the original study protocol. The central GCLLSG genetic reference testing laboratory in Ulm conducted FISH, mutation analysis of genes recurrently mutated in CLL (such as *TP53*, *ATM*, *NOTCH1*, and *SF3B1*) by targeted resequencing and IGHV mutation status, telomere length, GEP Exon- and SNP-array hybridization and analysis.

Trial participation, genetic testing, and data analysis have been conducted after informed patient consent, with the approval of the respective local ethics committees of participating centers. We have complied with all relevant ethical regulations and data analysis related to this study was approved by the Ulm University ethics committee.

**RNA isolation and quality assessment.** Ficoll density gradient centrifugation for isolation of mononuclear cells was performed on all CLL samples, immunomagnetic tumor cell enrichment via CD19 (Midi MACS, Miltenyi Biotec, Bergisch Gladbach, Germany) was performed on ($n = 337$) samples from the CLL8 trial, ($n = 300$) from the REACH trial and ($n = 5$) healthy donors (2 male, 3 female) (S1 sample set). Additional 89 samples were left unsorted (S2 sample set). For $n = 10$ CLL8 cases both the CD19 positive (+) and negative (−) compartments from CLL samples with inflammatory and non-inflammatory signatures were investigated. Only cases where the CD19 negative fraction had <3.30% (median 2.5%) contamination with CD19+ CD5+ cells were used. Total RNA for mRNA profiling was extracted from whole cell lysate according to the AllPrep DNA/RNA mini kit (Qiagen). Quality control was assessed using the Agilent 2100 Bioanalyzer with the RNA 6000 Nano LabChip (Agilent Technologies). The chip was prepared according to the manufacturer's protocol and analyzed using the 2100 Expert software (version 2.6). To ensure the best accuracy and reproducibility samples with an RNA integrity number (RIN) less than 7.0 were excluded from further analysis.

**Gene expression profiling on human exon 1.0 ST arrays or using quantitative polymerase chain reaction (qPCR).** Samples were analyzed for mRNA expression using the Affymetrix GeneChip Human Exon 1.0 ST Array (Affymetrix, Santa Clara, CA, USA). The experiment was conducted according to the manufacturer's protocol. In brief, 250 ng RNA per sample was amplified, transcribed to cDNA, fragmented, and subsequently labeled with biotin. Array hybridization was performed at 45 °C for 16–18 h in the Affymetrix GeneChip Hybridization Oven 640, arrays were subsequently washed in the Fluidics Station 450 and scanned on the GeneChip scanner 3000 7G. Complete microarray data sets are available at Gene Expression Omnibus (http://www.ncbi.nlm.nih.gov/geo/; GEO accession number: GSE58211 (REACH only); GSE126595 (full clinical data set); GSE126699 (functional data).

RNA extraction and expression analysis using qPCR: Total RNA was isolated using RNeasy or DNA/RNA AllPrep mini kit (Qiagen) as per the manufacturer's instructions. RNA concentration was estimated using NanoDrop (Thermo Scientific) and 400 ng of RNA was reverse transcribed using the Reverse Transcription Kit (Promega). The cDNA was diluted 1:10 prior to addition into the qPCR reaction mix. Sybr Green Supermix (Bio-rad) was used for the qPCR analysis as per the manufacturer′s protocol. Fold differences in gene expression were analyzed using the ΔΔ Ct method by normalizing to control sets as mentioned in the figure legends. Primers and sequences are provided in Table 1. The expression level of *miR-200c* (Cat. No. 002300) was analyzed using the TaqMan™ miRNA assays from ThermoFisher Scientific, according to the manufacturer's instruction. *U6* snRNA (Cat. No. 001973) was used as a control. In brief, total RNA was analyzed using the RNeasy kit from Qiagen and 10 ng of the RNA samples were reverse transcribed using the TaqMan™ MicroRNA Reverse Transcription Kit (Cat. No. 4366597) according to the manufacturer's instructions. The reverse-transcribed cDNA was diluted 1:3 and analyzed using the TaqMan™ Universal PCR Master Mix, no AmpErase™ UNG (Cat. No. 4364343). The qPCR reaction was performed in a total volume of 10 μl on a 384-well QuantStudio 5 Real-time PCR system from ThermoFisher Scientific.

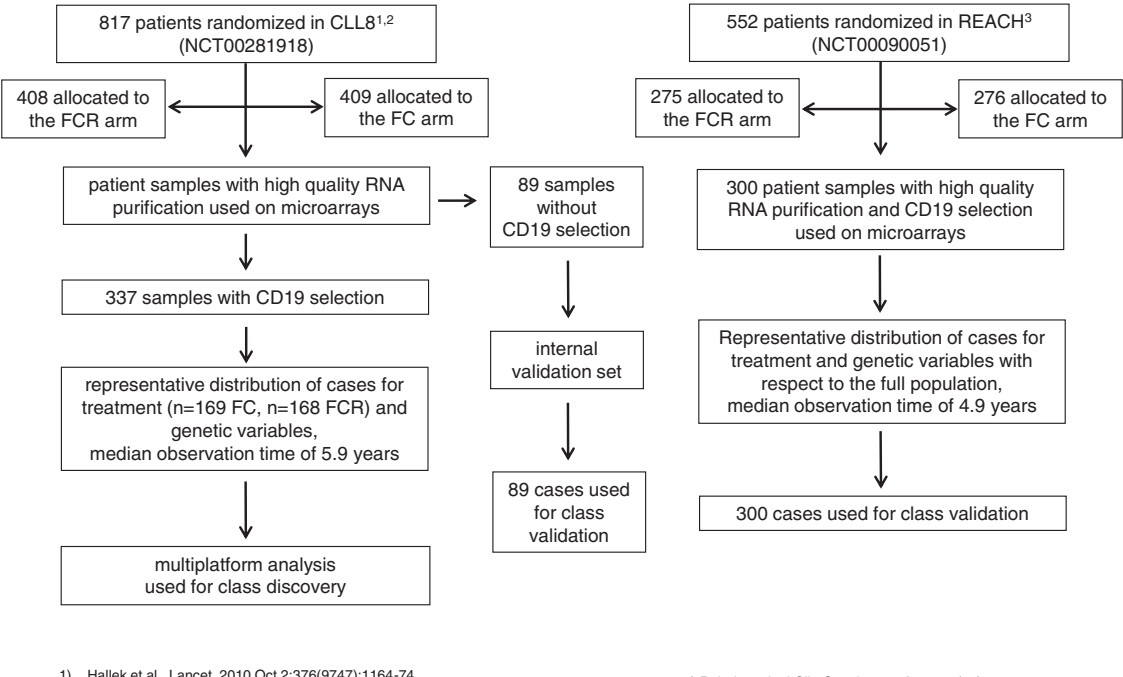

1) Hallek et al., Lancet. 2010 Oct 2;376(9747):1164-74
2) Fischer et al., Blood. 2016 Jan 14;127(2):208-15.

3) Robak et al., J Clin Oncol. 2010 Apr 1;28(10):1756-65

Abbreviations: FC = chemotherapy with fludarabine and cyclophosphamide, FCR = combined chemoimmunotherapy with FC plus rituximab

**Fig. 7 CONSORT diagram for the discovery and validation cohort.** CONSORT diagram providing information on enrollment and randomization for the CLL8 and REACH trial, along with details on the selection process of patient specimens used for class discovery and validation by gene expression profiling.

**Normalization of expression data**. Raw Affymetrix Human Exon 1.0 ST Array (HuEx-1_0-st-v2) data files and the data set from Herishanu et al. (GEO ID GSE21029) have been preprocessed by the robust multichip average (RMA) algorithm using the aroma.affymetrix R package[79] (version 2.12.0). Normalized data is stored with the assigned analysis ID, raw data files include info on CD19 selection and code of the CTG registry. Besides RMA normalization, background correction and quantile normalization were applied. Aroma.affymetrix was applied to generate gene expression values summarized on the exon/probe set level and on the transcript level using the "core" probe set definition according to Affymetrix. "Core" refers to probe sets that are supported by the most reliable evidence from RefSeq and full-length mRNA GenBank records containing complete CDS information. We further assessed and excluded the presence of potential batch effects induced by external factors including time point and location of sampling, duration of storage, and time point of labeling and hybridization. Quality control was further conducted with relative log expression and normalized unscaled standard errors, where abnormalities were not observed.

**Analysis of expression data**. Statistical procedures were performed with the R software environment, version 3.3.3 and 3.4.1. For GEP analysis BRB-ArrayTools Version 4.2.1–4.6.1 (available at http://linus.nci.nih.gov/BRB-ArrayTools.html and www.r-project.org) was used.

Unspecific filtering based on SD > 0.5 was used to select transcripts with the largest expression variability across all arrays for the CLL8 and REACH gene expression data set, resulting in 2359 transcripts (mRNA). Agglomerative hierarchical clustering and CC[80] on mRNA were applied using average or complete linkage and Pearson correlation distance metric with 1000 iterations, respectively. CC was used to identify the number of clusters with the best clinico-biologic segregation from $k = 2$ up to $k = 10$ possible clusters. The decision in favor of using $k = 6$ was based on combined information from the delta area plot, cluster stability, and clinical or biologic information as previously described. Differential expression, specifically assessed for genetic variables or defined clusters was conducted using the Class Comparison Tool from BRB-ArrayTools Version 4.2.1 with univariate permutation tests for individual genes controlling the false discovery rate (FDR) using the method of Benjamini and Hochberg. Genes for which the FDR was equal or less than 0.05 were considered significant for differential expression. Visualizing selected gene sets, such as components of a defined biological process, was conducted using the Genesis platform[81] (release 1.8.0). For depicting the cluster composition of selected gene sets hierarchical clustering was used with Pearson correlation distance and complete linkage when needed.

Specific assessment of differential expression for the most common recurrent alterations including *TP53* defect (del(17p) and/or *TP53* mutation), del(11q), tri(12), normal karyotype, del(13q), IGHV status, *SF3B1*, and *NOTCH1* mutations

were analyzed for respective groups defined by the presence of the cytogenetic alteration or mutation of interest, irrespective of the co-occurrence of other alterations. If not otherwise detailed, for group-specific assessments focusing on genetic categories like *TP53* defect, the group of interest was calculated against the reference group containing all other cases.

**Identification of biologic processes showing overrepresentation in expression clusters**. GSEA[82] was used to discriminate major biologic characteristics and processes in defined clusters (release v3.0). For the analysis, we used hallmark gene sets compiled at the Molecular Signatures Database, Broad Institute. GSEA was applied on mRNA expression data for respective cases of every single cluster in a comparative fashion with the remaining cases of all other clusters. Overrepresented gene sets of every single cluster with an FDR $q \leq 0.05$, which was used as stringent filter criteria, were selected for further analysis. Overrepresentation of biologic processes and resulting pattern composition for all clusters were visualized in a heatmap using the normalized enrichment scores of the overrepresented gene set as a measure for the enrichment intensity. Identified biologic processes were grouped together and labeled according to biologic similarities.

**SNP-array analysis**. SNP-array analysis on samples from the CLL8 trial for recurrent CNAs was conducted for $n = 309$ previously untreated samples of which $n = 18$ cases had a paired relapse sample[83]. All samples had GEP data with the respective cluster assignment available. In brief, for SNP–Array hybridization genomic DNA was hybridized to the Genome-Wide Human SNP Array 6.0 according to the manufacturer's protocol (Affymetrix, Santa Clara, CA, USA). SNP genotype calls were generated by applying the birdseed algorithm in Genotyping Console version 4.0 (Affymetrix) using at least 50 arrays in each analysis. DNA copy number analyses were performed using reference alignment[84], dChipSNP[85], and circular binary segmentation[86]. Segmentation was done pairwise against intra-individual reference DNA in cases having a pure CD19 negative cell fraction. For cases lacking matched normal material, the segmentation of each sample was computed against a pool of ten gender-matched reference samples. Resulting segments with a window of at least five consecutive markers and mean log2-ratios of >0.2 and <−0.2 were visually inspected using dChipSNP to exclude inherited copy number variants and false calls due to experimental artifacts (noise or interbatch effects). Lesions occurring in a subclone with a clone size under 25% were revised using the aroma.affymetrix software package[87] for an exact determination of segment boundaries. Size position and location of genes were identified with the UCSC Genome Browser; assembly March 2006, NCBI36/hg18 (http://www.genome.ucsc.edu/)[88]. DNACopy version 1.44.0 and dChip version 2010.01 were used. Microarray raw data contained in the analysis have been made publicly available at Gene Expression Omnibus (GEO accession number: GSE36908 (CLL8 treatment-naive) and GSE83566 (relapsed)). For visualization of deletion size for

**Table 1 Primers and TaqMan assays used for validation.**

Measurement of candidate genes involved in the EMT process for the serial TCL1A mouse transplant model

| Primer name | Species | Sequence |
|---|---|---|
| muVimentin_forward | Mouse | 5'CCAACCTTTTCTTCCCTGAAC |
| muVimentin_rev | Mouse | 5'TTGAGTGGGTGTCAACCAGA |
| muCDH1_forward | Mouse | 5'ATCCTCGCCCCTGCTGATT |
| muCDH1_reverse | Mouse | 5'ACCACCGTTCTCCTCCGTA |
| muSnai1_forward | Mouse | 5'CTTGTGTCTGCACGACCTGT |
| muSnai1_reverse | Mouse | 5'CAGGAGAATGGCTTCTCACC |
| muZEB1_forward | Mouse | 5'AGGTGATCCAGCCAAACG |
| muZEB1_reverse | Mouse | 5'GGTGGCGTGGAGTCAGAG |

Telomere measurement by quantitative PCR adapted from Richard Cawthon, Nucleic Acids Res. 2002 May 15;30(10):e47.

| Primer name | Species | Sequence |
|---|---|---|
| Telo std | Human | 5'-TTAGGGTTAGGGTTAGGGTTAGGGTTAGGGTTAGGGTTAGGGTTAGGGTTAGGGTTAGGGTTAGGGTTAGGGTTAGGGTTAGGG 3' |
| HB std | Human | 5'-GTCTGTGTGCTGGCCCATCACTTTGGCAAAGAATTCACCCCACCAGTGCAGG CTGCCTATCAGAAAGTGGTGGCTGGTGTGGC-3' |
| Tel 1b | Human | 5'-CGG TTT GTT TGG GTT TGG GTT TGG GTT TGG GTT TGG GTT-3' |
| Tel 2b | Human | 5'-GGC TTG CCT TAC CCT TAC CCT TAC CCT TAC CCT TAC CCT-3' |
| HBG3 | Human | 5'-TGT GCT GGC CCA TCA CTT TG-3' |
| HBG4 | Human | 5'-ACC AGC CAC CAC TTT CTG ATA GG-3' |

MiR-200c measurement

| TaqManTM miRNA assays from ThermoFisher Scientific | Species |
|---|---|
| miR-200c (Cat. No. 002300) | Human |
| U6 snRNA (Cat. No. 001973) | Human |

AICDA measurement

| TaqManTM assays from ThermoFisher Scientific | Species |
|---|---|
| AICDA: Hs00757808_m1 | Human |
| ACTB: Hs99999903_m1 | Human |
| 18S: Hs99999901_s1 | Human |

del(13q) LDBs the Integrative Genomics Viewer[89] (release 2.4.16) was used. Total number of CNAs (including non-recurrent CNAs) occurred with a mean count of 2.1 (range 0–9) per patient and showed homogenous distribution (range 1.7–2.2; GI: 2.2; EMT-L: 1.9; (I)EMT-L: 2; (I)GI: 1.7).

**Genomic identification of significant targets in cancer (GISTIC).** To assess the specific enrichment of genomic amplifications and deletions within clusters identified by CC of GEP, we applied GISTIC[18] (v2.0.23) to the curated SNP-array dataset. GISTIC identifies significantly amplified and deleted regions across a set of samples. Each aberration is given a $G$-score considering its amplitude and the frequency of its occurrence across samples within a GEP cluster. The significance of each aberration is estimated by GISTIC comparing the observed $G$-scores with results that would be expected by chance, using a permutation test that is based on the overall pattern of aberrations seen across the genome. To account for multiple testing, FDR estimation is done providing a consecutive $q$ value for each aberration. In our analysis, a $q$-value cut-off below 0.25 was considered to identify significant results (vertical green line in Fig. 2b).

**Longitudinal analysis for CNAs.** Longitudinal analysis for acquisition of CNAs before and after treatment was done on SNP-array data of cases with available baseline samples at inclusion into CLL8 trial (pre-treatment) and at the time point of relapse (post-treatment). Paired samples of cases with cluster assignment as identified through CC on GEP were available for GI baseline ($n = 11$), GI relapse ($n = 11$), (I)EMT-like baseline ($n = 7$), and (I)EMT-like relapse ($n = 7$). The Wilcoxon signed-rank test was applied to test for differences of aberrations in the pre- and post-treatment setting. Exemplary visualization for large representative CNAs in individual conditions was performed using dChip.

**Mutation analysis and signature projections for mutational processes.** Data were generated on samples from the CLL8 trial cohort. Cases used for this analysis were available for $n = 171$ matched cases with corresponding GEP and WES data[2]. Matched cases were distributed across identified clusters in representative numbers (EBF1-r $n = 6$, GI $n = 68$, EMT-L $n = 11$, (I)EMT-L $n = 52$, (I)GI $n = 31$, NRIP1 $n = 3$). Libraries for WES were constructed and sequenced on an Illumina HiSeq2000 or HiSeq2500 using 76 bp paired-end reads. For targeted sequencing we used Illumina Design Studio to create custom amplicons with a size of 250 bp covering all coding regions of TP53 and ATM. Library preparation was performed using TruSeq Custom Amplicon Assay Kit v1.5 (Illumina, San Diego, CA, USA) including extension and ligation steps between custom probes and adding of indices. Samples were pooled and loaded on a MiSeq flowcell in 48 sample batches

and sequenced with MiSeq Reagent Kit 500v2 (Illumina) for a paired-end run. Median depths of WES and targeted sequencing were 96× and 1332×, respectively. Software packages for bioinformatic analyses including demultiplexing, alignment to hg19 reference genome, variant calling, and annotation were used[2].

To assess the distribution and respective co-occurrence of enriched pathways and driver mutations, we used an agglomerative approach to estimate the similarity of distribution patterns for relative frequencies of mutations per cluster observed in our dataset, which was done in a hierarchical fashion (average linkage, Pearson correlation). Subsequently, a detailed representation of the single mutations was depicted for the respective clusters.

We used non-negative matrix factorization to assess the pathogenic processes operative in identified CLL subtypes which best explain the mutation pattern observed in corresponding cases as previously described[23]. Briefly, we used a fixed matrix of signatures which were hypothesized to be involved in aging, AID related DNA damage, and DNA repair deficiencies, as reported by Alexandrov et al.[23], to perform a projection of our data on to these signatures using the nonnegative matrix factorization multiplicative update as previously described[90]. After performing this projection, we used a heuristic approach to perform the signature selection. We selected the smallest set of signatures which produced a large drop in the cost function with respect to the signature sets with one more member or the same number of signatures but a different composition. This heuristic aims to perform parsimonious signature selection while still accurately representing the data. The applied SignatureAnalyzer algorithm is available at the Broad Institute homepage (https://software.broadinstitute.org/cancer/cga/msp).

**Protein extraction and western blotting.** Cluster-specific validation of protein expression levels was performed on samples from the respective cluster, not showing chromosomal aberrations other than del(13q) or gene mutations. For total protein extraction, cells were lysed in RIPA buffer (150 mM sodium chloride, 1% IGEPAL CA-630, 0.5% sodium deoxycholate, 0.1% sodium dodecyl sulfate (SDS), 50 mM Tris pH 8.0) supplemented with 1 mM DTT, 0.5 mM PMSF and phosphatase inhibitor cocktail, for 60 min at 4 °C. The amount of protein in each sample was quantified using the Protein Assay (Bio-rad). Equal concentrations of proteins were analyzed on 12% polyacrylamide gels or 4–12% Nu-Page pre-cast gels and subsequently transferred onto polyvinylidene difluoride (PVDF) membranes. The western blot images were acquired using the western gel documentation system. Antibodies used for western blot analysis in CLL cases not showing recurrent alterations include the following ones (catalog numbers are provided in brackets). From Cell Signaling: anti-AKT (#9272), anti-phospho-AKT(Thr308) (#4056), anti-phospho-p53 (ser15) (#9286), anti-PRMT5 (#2252). From Abcam: anti-c-Myc (#ab32072), anti-yH2AX (#ab26350), anti-mouse Alexa

Fluor 594 (#ab150116), anti-GAPDH (#ab8245). From Santa Cruz: anti-ERK1 (k-23) (#sc-94), anti-phospho-ERK(E-4) (#sc-7383), anti-RB(C-15) (#sc-50), anti-XPO1/CRM1 (H-300) (#sc-5595), anti-β-Actin (#sc-1615). From BD Bioscience: anti-p53(CM5) (#554293). From Thermo Fisher: HRP-conjugated anti-mouse (#A16072). Anti-ERK1 (k-23) (#sc-94) and anti-β-Actin (#sc-1615) were diluted 1:1000 in 5% BSA + TBST 0,1%, all other antibodies were diluted 1:500 in 5% BSA + TBST 0,1%, incubation was performed at 4 °C overnight. For image analysis of western blots, the intensities of individual bands in western blots were analyzed using Fiji ImageJ densitometry software (version 1.51j). The levels of the proteins were expressed relative to the loading controls (Actin). Phosphorylation levels of proteins were expressed as a relative measure compared to that of the total protein and their respective loading controls (Actin).

**Eμ-Myc/Eμ-TCL1 transgenic and BCL₁/Eμ-TCL1 transplantation mouse models**. *Ethics oversight*. We have complied with all relevant ethical regulations, all animal experiments were performed with the approval of the respective governmental authorities and local animal experimental ethics committees in each institution. The TCL1 serial transplant mouse model was performed according to protocols approved by the state government of Baden-Wuerttemberg, following the animal welfare guidelines (Registration 1124 and 1128), and was approved by the Ulm University animal experimental ethics committee. The BCL₁ syngeneic transplant model, Eμ-Myc, and Eμ-TCL1 mouse experiments were conducted under the Home Office licenses PPL30/2964 and P4D9C89EA following approval by local ethical committees, reporting to the Home Office Animal Welfare Ethical Review Board (AWERB) at the University of Southampton. Animals were maintained and bred in a pathogen-free environment (SPF IVC barrier) with a 14/10 day and night cycle, the temperature at 21 °C and humidity at 55%, as well as water and food ad libitum.

*BCL₁ syngeneic transplant model to validate induction of EMT-like networks in lymphoma*. To validate the potential for induction of an EMT-like program and corresponding dynamics in B cell lymphoma cells, we used a syngeneic BCL₁ tumor transplant model. For this $1 \times 10^5$ BCL₁ tumor cells were inoculated IV into Balb/c mice and spleens harvested at defined periods of time (day 7 ($n = 12$), 14 ($n = 6$), 17 ($n = 6$), and 21 ($n = 6$) alongside naive tumor-free mice ($n = 4$). Spleens were snap-frozen and then sectioned to provide material from which to extract RNA. Total RNA was isolated, assessed for purity using NanoDrop at 260/280 nm and the Bioanalyser. Samples with RIN scores >7 were taken forward and subjected to RNA sequencing (EA²).

*Eμ-TCL1 serial transplant mouse model to assess EMT-like plasticity in lymphoma*. The Eμ-TCL1 tumors prior to the start of the experiment were transferred and expanded once in syngeneic C57BL6/J mice. For serial transfers, 10 million splenic tumor cells were transplanted by intravenous injection and the animals were sacrificed when the mice appeared critically sick, a surrogate endpoint that was defined based on scoring for disease severity including WBC count, changes in mobility, signs of suffering, as approved by the local animal experimental ethics committee. The tumor cells were purified with ficoll and only tumors with more than 90% CD5⁺ CD19⁺ CLL cells were used for serial transfers and analyses. Three rounds of serial transfers were performed and the tumors were isolated from the spleen. The EMT markers Vimentin (*Vim*), Cadherin-1 (*Cdh1*), and corresponding transcription factors *Zeb1* and *Snai1* were measured using qPCR on tumors isolated from the different serial transfers.

*Eμ-Myc/Eμ-TCL1 mouse model and proteome profiling to validate GI-specific networks and associated inhibition of EMT-like networks*. Mass spectrometry (MS) proteomics analyses of Eμ-Myc and Eμ-TCL1 tumors were performed as described[91] and submitted to GSEA. Spontaneous tumors from female Eμ-Myc [C57BL/6J-TgN(Ighmyc)22Bri/J] hemizygous and Eμ-TCL1 [C57BL/6J-TgN(IghTCL1)22Bri/J] hemizygous mice were compared with wild-type controls aged 6 weeks and 200 days, in addition to the pre-terminal model, controls taken at 6 weeks of age. B cells were isolated from spleens by magnetic isolation kit (Miltenyi Biotech, Bergisch Gladbach, Germany) and snap-frozen. Samples were pooled with four tumors from each model assigned to two pools of two tumors and non-tumor pools of six samples, to be accommodated in a single eight-plex. Snap frozen cell pellets were lysed in 0.5 M TEAB with 0.05% SDS, with 100 μg of protein lysate per pool TCEP-reduced, MMTS-alkylated, trypsin digested, and labeled with isobaric tags for relative and absolute quantitation (iTRAQ) 8-plex according to the manufactures instructions (ABSciex, Framingham, MA). Labeled peptides were combined and pre-fractionated using a 90-min high-pH reverse-phase C8 fractionation (2–30% organic) collecting 69 peak-dependent fractions. Each fraction was analyzed by LC–MS/MS (Dionex Ultimate 3000 and Orbitrap Elite (Thermo Scientific)) over 200 h of MS time using top 12 data-dependent acquisition and 120,000 resolution with reporter ions captured with HCD at 35 keV at 15,000 resolution. Raw data were analyzed by Proteome Discoverer 1.4.1.14 with SequestHT 1.1.1.11 and Percolator modules searched against the mouse UniProt Swissprot and trembl databases (downloaded 01/15). The raw data and processed outputs are available at https://www.ebi.ac.uk/pride/archive/projects/PXD004608. Relative expression was assigned from iTRAQ reporter regions and was median normalized and quality-adjusted using spiquetool.com. Log₂ (ratios) were generated describing each tumor sample pool relative to the two WT control pools, in addition to the 6-week pre-terminal model controls relative to the 6-week WT control. A value summarizing all 4 log₂ (ratios) for each tumor model was also used

(mean/(SD + 1)). Lists of gene names and corresponding log₂ (ratios) and summary values for all 8270 proteins were analyzed by GSEA 3.0 using the GSEA-preranked approach due to a non-standard data format enriching for the MSigDB H and C2 gene sets. All default settings were used.

**Radiation-induced DNA damage**. The human B cell lines, MEC1, MEC2, JVM2, JVM3, LCL-WEI, EHEB, and Granta were purchased from the German Collection of Microorganisms and Cell culture (Deutsche Sammlung von Mikroorganismen und Zellkulturen, DSMZ) with the certificate from the vendor and were additionally authenticated through sequencing by Multiplexion GmbH. Cell lines were cultured in RPMI medium with 10% FCS and 1% L-glutamine/were maintained in IMDM medium with 10% FCS and 1% L-glutamine. All cells were tested for mycoplasma contamination monthly. For induction of DNA damage, the cells were irradiated with 5 Gy γ-irradiation. The cells were collected 4, 8, 16, 24, and 48 h after ionizing irradiation with 5 Gy, and expression changes in *miR-200c*, *TP53*, *TP63*, *ATM*, *ZEB1*, and *TWIST1* were analyzed for individual time points and in comparison to the corresponding nonirradiated sample.

**Assessment on AID induced genomic instability**. *Cell culture*. BL2 cell line was obtained from the German Collection of Microorganisms and Cell culture (ACC 625. Deutsche Sammlung von Mikroorganismen und Zellkulturen, DSMZ). BL2 *AICDA-* cells were kindly provided by Claude-Agnes Reynaud (INSERM U1151, Paris) and were described previously[92]. Human embryonic kidney HEK293T were obtained from the European Collection of Authenticated Cell Cultures (ECACC) (Culture Collections, Public Health England, Salisbury, UK). Cell lines were authenticated by DSMZ (https://www.dsmz.de/collection/catalog/human-and-animal-cell-lines/identity-control) or Public Health England (https://www.phe-culturecollections.org.uk/media/153328/ccw5704-culture-collections-quality-policy.pdf). BL2 *AICDA-* cells were not authenticated. Cells were cultured in RPMI 1640 (BL2) or Dulbecco's Modified Eagle Medium (HEK293T) (Sigma-Aldrich, Dorset, UK) supplemented with 10% (v/v) fetal bovine serum, 100 U/ml penicillin, 100 U/ml streptomycins at 37 °C in a humidified atmosphere containing 5% CO₂. All cells were tested for mycoplasma contamination monthly. Lentiviral transfection: Packaging transfection was performed in HEK293 cells using Lipofectamine® LTX Reagent (Invitrogen®, Carlsbad, CA, USA) and the following vectors: the pRSV-Rev packaging, pMDLg packaging, pMD2.G envelope, and pLenti-C-mGFP vector expressing *AICDA* (NM_020661) Human Tagged ORF Clone (RC202949L2, OriGene, Cambridge, UK). Lentivirus-containing medium was harvested 24 or 48 h post-transfection, filtered through 0.45 μm PES filter, and concentrated using Retro-X Concentrator (Clontech, Takara Bio, USA). After overnight incubation at +4 °C, the virus-containing mixture was centrifuged for 45 min at 500*g* and the pellet was resuspended with the medium in 1:10 of the original volume. Totally, 107 of BL2 *AICDA-* or HEK 293T cells were then exposed to a virus-containing medium for 24 h. Infection efficiency was then determined by flow cytometry as a percentage of GFP+ cells. AID-GFP cells were then FACS sorted using BD FACSAria II Cell Sorter (BD Biosciences, Wokingham, UK). Immunofluorescence: For immunofluorescence staining, cells were harvested and cytospun onto microscope slides. Slides were fixed in −20 °C methanol and washed in TBS/0.05% Tween. Primary mouse anti-γH2AX (ab26350, Abcam, Cambridge, UK) antibody, diluted 1:500 in TBS/0.1% BSA, was then applied for 1 h at room temperature. After three washing steps with TBS/0.05%Tween, slides were stained with secondary anti-mouse Alexa Fluor 594 (ab150116, Abcam, Cambridge, UK), diluted 1:500, for 1 h at room temperature.

Following three times with TBS/0.05% Tween. Nuclei were counterstained with DAPI and slides were mounted in ProLong Gold Antifade Reagent (Invitrogen, Life Technologies, Paisley, UK). Images were taken using Nikon Ci-L upright fluorescence microscope and Nikon NIS Elements AR software (Ver4.30.01, 64-bit edition). Assessment of γ-H2AX was used as a universal marker of DNA damage, including DNA double-strand breaks[93]. Western blot: Total protein lysates were mixed with NuPAGE® LDS Sample Buffer (Thermo Fisher Scientific, Paisley, UK), and then 10 μg of protein were resolved on precast 4–12% Bis–Tris Protein Gel (Thermo Fisher Scientific, Paisley UK). Proteins were transferred by wet transfer onto Immobilon-P Membrane PVDF membrane (Merck Millipore, Billerica, Massachusetts, USA), blocked with 5% non-fat skim milk in TBS, and then the membrane was incubated with appropriately diluted primary antibodies for one hour at room temperature in TBS/0.1% Tween. After three washing steps, membranes were incubated with HRP-conjugated anti-mouse antibody (1:2000, A16072, Thermo Fisher, Paisley, UK). After three washing steps, membrane chemiluminescence was analyzed by Amersham ECL Prime Western Blotting Detection Reagent (GE Healthcare®, Little Chalfont, UK) and ChemiDoc imaging system (BioRad, Hemel Hempstead, UK). The following primary antibodies were used: mouse anti-γ-H2AX [9F3] (1:1500, ab26350, Abcam, Cambridge, UK), mouse monoclonal [6C5] to GAPDH (1:3000, ab8245, Abcam, Cambridge, UK). Sister chromatid exchange assay: we used the sister chromatid exchange (SCE) assay to assess cellular genotoxicity and ongoing mutagenesis[94,95]. Cells were cultured in DMEM containing 10 μM 5-bromo-2′-deoxyuridine BrdU (ab142567 Abcam, Cambridge, UK) for two cell divisions cycles. After 4 h treatment with Colcemid (0.02 μg/mL, 10295892001, Sigma-Aldrich, Gillingham, UK), cells were incubated with prewarmed 75 mM KCl solution for 20 min at 37 °C. Then, cells were spun down and fixed using Carnoy's fixative (methanol: glacial acetic acid).

Mitotic cells were dropped on pre-chilled microscope slides and left to dry in the dark at room temperature. Slides were immersed in acridine orange (A1301, Thermo Fisher Scientific, Paisley, UK) for 5 min, mounted in 2× SSC buffer, and covered with a coverslip. Images of chromosome spreads were obtained using a Nikon Ci-L upright fluorescence microscope and Nikon NIS Elements AR software (Ver4.30.01, 64-bit edition). Sister chromatid exchanges were quantified microscopically from at least five random fields containing at least 20 metaphase spreads. Alkaline Single Cell Electrophoresis (Comet) assay: Single Cell Electrophoresis was performed using CometAssay Kit (4250-050-K, Trevigen, AMS Biotechnology, Abingdon, UK) as per manufacturer's instructions. Briefly, 105 cells were harvested by centrifugation and then resuspended in 1X PBS, mixed with 0.5% low melting agarose, and plated onto comet slides pre-coated with normal melting point agarose. Subsequently, slides were immersed in comet lysis solution for one hour at 4 °C. Slides were then further treated with alkaline solution (NaOH pH 13, 200 mM EDTA in H2O) submerging in an electrophoresis tank. Alkaline electrophoresis was performed for 30–60 min at 21 V (300 mA). Nuclei and "comet tails" were subsequently stained with SYBR® Gold stain (Thermo Fisher Scientific, Paisley, UK). Slides were visualized using fluorescence microscopy, and % of tail DNA on per cell basis was determined using an open-source Cell Profiler (https://cellprofiler.org/) software equipped with the Comet Assay analysis module (https://cellprofiler-examples.s3.amazonaws.com/ExampleCometAssay.zip). Triplicate slides were processed per each *AICDA*-related condition with at least 70 comets analyzed per each condition.

**Telomere length analysis**. The telomere length measurement was carried out using a qPCR–based technique[96]. The primers used to amplify telomere and single-copy genes (SCG) were tel1b, tel2b, and HBG3, HBG4, respectively[96]. The absolute telomere length was obtained by using synthetic oligonucleotide standards for telomere (84 bp) and SCG (81 bp) PCR. Briefly, a tenfold dilution of the telomere and SCG standard was prepared and the number of DNA molecules in each standard was calculated as described[97]. Twelve nanograms of DNA was used per reaction (total volume of 10 μl) in triplicates for the telomere and SCG PCRs and amplified using Qiagen quantitect SYBR green in 384-well plates and analyzed using 7900HT fast real-time PCR system (Applied Biosystems). Six telomere length controls with known telomere length, analyzed using terminal restriction fragment length analysis (TRF) were included in every plate to detect variations. The qPCR technique was validated by TRF length analysis and Southern hybridization. Totally, 6 μg of non-degraded DNA was digested overnight using Hinf I and Rsa I and resolved on a 0.8% agarose gel. In-gel hybridization was carried out by drying the gel and hybridizing with a telomere-specific probe, end-labeled with $^{32}$P. The mean telomere length was analyzed from the autoradiograph. A correlation of $R^2 = 0.8516$ was obtained upon a comparison of telomere length measured using qPCR and TRF, in a control sample set ($n = 18$). The TRF value of telomere length for each sample was calculated from the linear regression of qPCR vs. TRF.

**RRBS and methylation analysis**. Genomic DNA from $n = 182$ matched CLL8 samples was used to produce RRBS libraries. They were generated by digesting genomic DNA with MspI to enrich for CpG-rich fragments and then ligated to barcoded TruSeq adapters (Illumina) to allow immediate subsequent pooling. It was followed by bisulfite conversion and PCR, as previously described[98]. Libraries were sequenced and aligned to the bisulfite-converted hg19 reference genome using Bismark (RRID: SCR_005604) v0.15.0[99].

*Methylation analysis*. Only CpGs with ten or more reads were included in the analysis. Promoters were defined as the regions encompassing 2 kb upstream and downstream of the transcription start site of UCSC genes. Promoters or genes with at least five covered CpGs were included in the analysis. Promoter or gene methylation was calculated by the average methylation levels of all the CpGs inside.

**Survival analysis**. CLL8 and REACH clinical trial data were analyzed on an intention-to-treat basis (eligible subjects were analyzed as randomized). Progression-free survival (PFS) was defined as the time from randomization to disease progression or death, overall survival (OS) was defined as the time between randomization and death. PFS and OS were estimated by the Kaplan–Meier method, and differences between groups were assessed using two-sided non-stratified log-rank tests. In addition, hazard ratios (HR) and 95% confidence intervals were calculated using Cox regression modeling. With regard to OS and PFS multivariable Cox proportional hazards, regression models were used to assess the independent prognostic value of identified major subtypes (GI, (I)GI, EMT-L, (I)EMT-L) in CLL8 and REACH. Additional prognostic factors in the models were treatment, *TP53* mutation, IGHV mutation, 17p deletion, 11q deletion, trisomy 12q, and 13q deletion.

**Statistical software**. Statistical analysis was performed with R version 3.3.3 and 3.4.1, with R package survival, version 2.41-2; SPSS version 24–26 (IBM, NYC, NY); Prism software version 6.0 h (GraphPad), MATLAB 2018b and BRB-ArrayTools Version 4.2.1–4.6.1.

**Reporting summary**. Further information on experimental design is available in the Nature Research Reporting Summary linked to this article.

## Data availability

Complete data sets are available: For GEP at Gene Expression Omnibus (http://www.ncbi.nlm.nih.gov/geo/; GEO accession number: GSE58211 (REACH only); GSE126595 (full clinical data set); GSE126699 (including functional data). For SNP-Microarray raw data at Gene Expression Omnibus (GEO accession number: GSE36908 (CLL8 treatment-naive) and GSE83566 (relapsed)). CLL8 WES data are deposited in dbGaP under accession code phs000922.v1.p1. CLL8 RRBS sequencing data are available from the NCBI (GEO accession number: GSE143673). The proteome profiling raw data and processed outputs are available at https://www.ebi.ac.uk/pride/archive/projects/PXD004608. All other relevant data supporting the key findings of this study are available within the article, its supplementary information, source data files, or from the corresponding first author upon reasonable request. A reporting summary for this article is available as a supplementary information file. Source data are provided with this paper.

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

## Acknowledgements

The authors thank all patients and their physicians for trial participation and donation of samples; the DCLLSG; Sabrina Schrell and Christina Galler for their excellent technical assistance. This work was supported by research grants from the Else Kröner-Fresenius-Stiftung (2010_Kolleg24, 2012_A146), BMBF (PRECISE), Cancer Research UK (A24721 and A18087), Kay Kendall Leukemia Fund (KKL1101), and Blood Cancer UK (19001), European Commission/BMBF ("FIRE CLL", 01KT160), SFB 1074 (Projects B1, B2), DJCLS R 11/01.

## Author contributions

J.B., D.M., and S.S. planned and coordinated multi-platform profiling. J.B. conceptualized the study, collected data, performed experiments, analyzed experimental and clinical data and established models. A.Br., A.T-W., B.M.C.J., J.K., H.P., G.A., K.H., A.S., H.E.J., L.N.D., M.J.C., O.E., D.A.L., D.S.N., M.S.C., and A.B. performed the experiments or analyzed data. S.R., A.G., J.K., A.B., and D.S.N. analyzed the clinical data. R-F.Y., B.E., L.B., K.F., E.T., M.W., T.R., and C.S. collected the patient samples or data. J.G., M.H., C.J.W., and H.D. provided conceptual inputs. A.Br. and T.K. designed and completed functional in vitro studies for AICDA. J.B. wrote the paper, with input from A.Br., M.S.C., A.B., D.S.N., D.M., and S.S. All authors reviewed and approved the paper.

## Funding

## Competing interests

S.S. received advisory board honoraria, research support, travel support, speaker fees from AbbVie, Amgen, AstraZeneca, Celgene, Gilead, GSK, Hoffmann-La Roche, Janssen, Novartis, Sunesis. O.E. is supported by Janssen, Johnson and Johnson, Volastra Therapeutics, AstraZeneca, and Eli Lilly research grants. He is a scientific advisor and equity holder in Freenome, Owkin, Volastra Therapeutics and One Three Biotech and paid scientific advisor to Champion Oncology. M.S.C. is a retained consultant for BioInvent International and has performed educational and advisory roles for Roche, Boehringer Ingelheim, Baxalta, Merck KGaA, and GLG. He has received research funding from Bioinvent, Roche, Gilead, iTeos, UCB, and GSK. He is co-inventor of patent WO2012022985A1 protecting antibodies directed to hFcgRIIB in combination with CD20 specific antibodies. L.B.: Advisory Committees Abbvie, Amgen, Astellas, Bristol-Myers Squibb, Celgene, Daiichi Sankyo, Gilead, Hexal, Janssen, Jazz Pharmaceuticals, Menarini, Novartis, Pfizer, Sanofi, Seattle Genetics. R-F.Y. and M.W. are employed by Genentech and Roche, respectively. J.B. received travel support from Janssen and research support from Roche. The remaining authors declare no competing interests.

## Additional information

[1]Department of Internal Medicine III, University of Ulm, Ulm, Germany. [2]Centre for Haemato-Oncology, Barts Cancer Institute, Queen Mary University of London, London, UK. [3]Broad Institute of Harvard and MIT, Cambridge, MA, USA. [4]Department I for Internal Medicine and Centre for

Integrated Oncology, University of Cologne, Cologne, Germany. [5]Division of Biostatistics, German Cancer Research Center, Heidelberg, Germany. [6]Caryl and Israel Englander Institute for Precision Medicine, Weill Cornell Medicine, New York, NY, USA. [7]Department of Physiology and Biophysics, Weill Cornell Medicine, New York, NY, USA. [8]Institute for Computational Biomedicine, Weill Cornell Medicine, New York, NY, USA. [9]Genomics Core Facility, Ulm University, Ulm, Germany. [10]Centre for Cancer Immunology, Cancer Sciences, Faculty of Medicine, Cancer Research UK Centre and Experimental Cancer Medicine Centre, University of Southampton, Southampton, UK. [11]Biostatistics, Genentech Inc., South San Francisco, CA, USA. [12]Medical Clinic for Hematology, Oncology and Tumor Biology, Charité University Hospital, Berlin, Germany. [13]Roche Pharma Research and Early Development, Penzberg, Germany. [14]Department of Hematology, Medical University of Lodz, Lodz, Poland. [15]Department of Pharmacology and Therapeutics, Faculty of Life and Health Sciences, Institute of Systems, Molecular and Integrative Biology, University of Liverpool, Liverpool, UK. [16]Sandra and Edward Meyer Cancer Center, Weill Cornell Medicine, New York, NY, USA. [17]Cancer Genomics and Evolutionary Dynamics, Weill Cornell Medicine, New York, NY, USA. [18]New York Genome Center, New York, NY, USA. [19]Biostatistics and Computational Biology, Dana-Farber Cancer Institute, Boston, MA, USA. [20]Department of Medical Oncology, Dana-Farber Cancer Institute, Boston, MA, USA. [21]Department of Internal Medicine, Brigham and Women's Hospital, Boston, MA, USA. [22]Harvard Medical School, Boston, MA, USA. [23]German Cancer Research Center (DKFZ), Heidelberg, Germany. ✉email: johannes.bloehdorn@gmail.com; d.mertens@dkfz-heidelberg.de

