## [Peer Review File · Nature Communications]

Multi-platform profiling characterizes molecular subgroups and resistance networks in chronic lymphocytic leukemiaREVIEWER COMMENTS

Reviewer #1 (Remarks to the Author):

This manuscript describes how samples from patients with CLL in need of treatment can be subcategorised into a number of different subgroups based on GEP consensus clustering, GSEA and incorporation of additional layers of information from arrays and sequencing and conventional diagnostics.

Initially, 6 clusters are identified. The majority of the subsequent work is then performed on GI/GI-I and EMT-L/I or non-I vs I groupings.

Although the subgroups are genomically distinct and associate with known aberrations such as mutation signatures and DDR gene mutations, they cut across IgHV status and TP53 aberrations and therefore could represent novel findings.

Some biological data in murine models is presented in an attempt to unpick the interplay between DDR and inflammation.

Findings are then correlated to clinical outcomes and validated in an independent clinical cohort.

The study has got several strengths, but also significant weaknesses that need to be addressed:

Strengths: analyses are performed on large datasets from uniformly treated patients with in clinical trials. The clinical outcome data is therefore robust. Although chemoimmunotherapy is no longer SOC of patients with CLL, any molecular subgroupings that predict outcome from these historical data sets with long follow-up is still highly informative.

The authors validate their findings in an independent group of patients receiving relapsed therapy.

They used unsorted cells as a control for the robustness of their GEP-based approach and also to identify unique GEP in the CD19neg fractions.

Major weaknesses

1. There was a 50% dropout in sample availability. This is likely because of the high number of tests performed and the fact that the samples came from multicentre studies. This should be highlighted and discussed in the discussion. For example, it is possible that the study includes a bias towards samples with higher absolute lymphocyte count.

2. Related to this: The approach taken required 7 different test modalities and would never be applicable in the clinic. Therefore, please, remove any suggestions that this approach might be clinically translatable.

3. It is not clear that the clusters could not have been derived simply from DNA analyses: For example, for the GI/I-GI cluster it is stated that “cases with TP53 inactivation, V3-21 usage, short telomeres, high white blood cell (WBC) counts or ZAP-70 positivity ($p < 0.05$, Fisher’s exact test) were enriched in GI/(I)GI clusters. Furthermore, genes involved in maintenance of genomic stability were frequently mutated. Both GI and (I)GI involved frequent gains of 8q24.21 (including MYC) and 2p16.1 (including XPO1, REL). GI further showed gains of 6q22.31 and losses of 15q15.1 (including KNSTRN, BUB1B), 10q24.3 and 6q21. (I)GI showed losses of 13q14.13. Amplifications of MYC (8q24.21) were most frequently observed in GI/(I)GI cases. Moreover, IGHV mutated cases showed significantly higher activation of signatures 15 ($p = 0.01$), 3 and 20 ($p < 0.005$) (Mann-Whitney), indicating defective DNA repair.”

The authors should use the combination of the different DNA-based result to demonstrate that these are insufficient to generate precise clusters and that it requires the GEP in order to derive prognostically significant subgroups.

4. The authors derive an immune signature that they then try to further validate in murine models.

Unfortunately, I really cannot follow the result section relating to this and figure 4.

Please, include a rationale for the choice of murine models and why precisely these models were used to either unpick the relationship between genomic instability and EMT stimuli and/or were chosen to model the specific disease context of CLL.

Please, summarise the experimental design and what you did at each step and why. This has to be summarised in each figure concisely and clearly. Please, focus on the key experiments only in the figure 4.

5. Regarding the epigenetic modification that might shape pathogenic networks and subgroupings: this claim is currently purely based on gene expression results. I would therefore advise to either remove the entire section or to produce supporting CHIP-Seq or ATAC-Seq data from a couple of key genes to show that key network genes are reprogrammed as a result of epigenetic modification.

6. Cases without TP53 defect showed PFS rates at 5 years of 17% in GI vs. 47% in (I)EMT-L (GI: median PFS 29.8 vs. (I)EMT-L: 39.5 months, HR:1.83 (95%CI 1.12-3.0), p=0.016) when treated with FC, but was the same for GI and EMT-L with the addition of rituximab.

In the CLL8 study, this shift of the PFS curve with the addition of Rituximab was also observed in patients with del11q. It is important to show whether the shift of GI reflects patients with these and/or other DDR abnormalities who are TP53 wild-type, in particular also patients with bi-allelic ATM mutations.

Conversely, in CLL8, a lack of benefit from the addition of rituximab was described for patients with NOTCH1 abnormalities. These are enriched for in the EMT-L group that equally lacks this improvement.

More generally, it is important to understand that the prognostic relevance of the GI and EMT subgroups is independent and due to the specific GEP and NOT because of their association with ATM and other genomic abnormalities.

Related to this, PFS curves of the conventional markers TP53, IgHV, del11q, Notch1 and SF3BI should be shown as a comparator to investigate significant differences between these and the novel subgroupings based on GEP.

Reviewer #2 (Remarks to the Author):

- Throughout figure 1, were all samples used, or just CLL8 samples. It looks like the initial characterization in Figure 1A might have been done in CLL8 and then validated with the REACH samples, but I am not sure if this is the case. If this is, Were the EBF1 and NRIP1 clusters not found in the REACH samples?
- Was any validation of the GEP data performed on fresh cells? Both of the trials chosen have very old samples, and I think it would be important to make sure there is no freeze/thaw artifact. As well, were distinct differences found between the trials or between specific sites that initially processed the samples?
- It would be helpful to see the specific genes that define the subgroups in figure 1b
- Were the FISH, telomere length, TP53, zap-70, etc performed in a centralized manner for this study?
- It may be worth noting that the GI groups seemed much more likely to have TP53 mutation without del17p than the EMT groups (if this is actually significantly different; it does look so visually)
- It is interesting that del11q frequency does not appear different among the clusters. It also looks like del11q appears more frequently than predicted (usually seen in about 20% of patients in frontline trials, visually looks like more than that). Do you think that 11q + another abnormality might be more important for GI or DDR abnormalities than del11q as a single abnormality?
- Since prognostically del13q is more relevant as a positive biomarker when it exists as a sole abnormality, I wonder if any of the analyses of chromosome abnormalities were performed separating out samples with del13q alone?
- Figure 2B is a little confusing for me to interpret, as I don't understand what the numbers at the top

of the individual panels mean (% of samples? Clonal fraction?), and why is the scale and numbers at the top different for each of the panels. I also don't understand the interpretation that myc abnormalities are seen more frequently in the GI groups, when visually the (I)EMT group also seems to have a line in the same area as Myc in the GI groups. Also, the (I) EMT cluster seems to have a number of chromosome gains that would be interesting to know what those areas represent.

- In Figure 3, I am not sure what the biologic relevance of total p53 or phospho p53 being elevated in the GI groups. While they are different, by itself I don't know that these basal levels are functionally relevant. Perhaps it would be of greater interest to show p53 induction after ionizing radiation.
- Why do the survival curves in Figure 5 only include (I) EMT and not the non-inflammatory cluster?

Ulm 15.11.2020

Response to referees letter

Manuscript: "Multi-platform profiling characterizes molecular subgroups and resistance networks in chronic lymphocytic leukemia"

Manuscript submission no.: NCOMMS-20-13007

We would like to thank the reviewers for their constructive comments and criticism that helped us substantially improve the quality of our manuscript. We have thoroughly revised the manuscript in order to address all the reviewers' concerns.

Together with the revised manuscript, we submit below a point-by-point response explaining the actions taken to address each and every one of the comments and suggestions we received. We hope that you will find the revision thorough and satisfactory.

General changes to the manuscript:

Changes of word count and references (previous to submission (black) and after revision (red)):

Word count abstract: 159 - 155 after revision

Word count text: 4281 - 5082 after revision

Figures: 5 - 6 after revision

Supplements: Figures: 7; Tables: 8 - Figures 14, Tables 10 after revision

References: 73

Specific changes according to reviewers comments:

Reviewer #1 (Remarks to the Author): This manuscript describes how samples from patients with CLL in need of treatment can be subcategorised into a number of different subgroups based on GEP consensus clustering, GSEA and incorporation of additional layers of information from arrays and sequencing and conventional diagnostics. Initially, 6 clusters are identified. The majority of the subsequent work is then performed on GI/GI-I and EMT-L/I or non-I vs I groupings. Although the subgroups are genomically distinct and associate with known aberrations such as mutation signatures and DDR gene mutations, they cut across IgHV status and TP53 aberrations and therefore could represent novel findings. Some biological data in murine models is presented in an attempt to unpick the interplay between DDR and inflammation. Findings are then correlated to clinical outcomes and validated in an independent clinical cohort. The study has got several strengths, but also significant weaknesses that need to be addressed: Strengths: analyses are performed on large datasets from uniformly treated patients with in clinical trials. The clinical outcome data is therefore robust. Although chemoimmunotherapy is no longer SOC of patients with CLL, any molecular subgroupings that predict outcome from these historical data sets with long follow-up is still highly informative. The authors validate their findings in an independent group of patients receiving relapsed therapy. They used unsorted cells as a control for the robustness of their GEP-based approach and also to identify unique GEP in the CD19neg fractions.

Response: We thank the reviewer for recognizing the importance of the questions addressed in this work as well as its comprehensiveness.

Major weaknesses

1. There was a 50% dropout in sample availability. This is likely because of the high number of tests performed and the fact that the samples came from multicenter studies. This should be highlighted and discussed in the discussion. For example, it is possible that the study includes a bias towards samples with higher absolute lymphocyte count.

Response: We thank the reviewer for this constructive comment and fully agree on the importance of providing further details. As correctly assumed material availability was limited due to the multiple tests performed on single specimens and varying amount of material available from submitting centers.

Material chosen for GEP analysis in the discovery cohort was further selected based on availability of CD19+ sorted cells and stringent quality control. To ensure the best accuracy and reproducibility of GEP results samples with a RNA Integrity Number (RIN) less than 7.0 were excluded from further analysis, additionally reducing available material.

To provide further information on the characteristics of patients from whom material was used for GEP and additional analyses, we have extended the information provided in supplementary table 1 which now contains information on patient characteristics for the full CLL8 trial cohort (n=817), the GEP discovery cohort of CD19+ sorted cases (n=337) and the patients where CD19+ samples for GEP were not available (n=480).

For the vast majority of characteristics there are only small differences regarding the CD19+ sorted GEP cohort compared to the full trial population, with overall differences affecting only few variables, which do not impact the results generated from the multiplatform profiling approach.

Specific changes made to the manuscript:

Extended information in table S1.

Methods summary, lines 142-143: “Multiparameter analysis was conducted for CD19 sorted CLL8 samples and distribution of genetic characteristics for analyzed cases was representative for the full CLL8 trial cohort (table S1).”

Discussion, lines 519-522: “While we observed higher leukocyte counts for the CLL8 discovery cohort of CD19 sorted CLL cases, likely through selection of samples with abundant material for multiple analyses, patient characteristics and especially high-risk markers showed a well-balanced distribution representative of the full trial population.”

2. Related to this: The approach taken required 7 different test modalities and would never be applicable in the clinic. Therefore, please, remove any suggestions that this approach might be clinically translatable.

Response: We fully agree that performing these 7 respective analyses for subtype identification in the clinical setting is not viable. Rather the approach was taken to identify the key subgroups with subsequent validation.

For clinical application we intended to provide potential opportunities to improve treatment efficacy in a given biologic context by exploiting dependencies or synergistic effects, e.g. through novel treatments in development. As such we have observed subtype specific pathway enrichment or overexpression of biologic targets like XPO1, BCL2, PRMT5, PRMT1, EZH2 etc. for which novel inhibitors are in development or recently approved.

Specific changes made to the manuscript:

Abstract, lines 77-78:

Original: “This work provides a novel perspective on CLL biology and risk categories in *TP53* wild-type CLL. Molecular targets within subgroups may advance personalized treatment approaches in CLL.”

Now: “This work provides a novel perspective on CLL biology and risk categories in *TP53* wild-type CLL. Further, the identified molecular targets identified within each subgroup provide opportunities for new treatment approaches.”

Introduction, lines 95-96:

Original: “However, the context in which genetic alterations arise remains to be further explored to understand disease dynamics and refine personalized treatment options.”

Now: “However, the context in which genetic alterations arise remains to be further explored to understand disease dynamics and refine therapeutic strategies by targeting cellular network or genetic dependencies.”

Discussion, lines 609-617:

Original: “This study extends the basis for understanding CLL pathogenesis and pathway dependencies that may be targeted by novel treatment approaches, including inhibitors targeting exportins or protein methyltransferases. Future assessment of the subtype related outcome in trial cohorts testing BCL2-, BTK- and other inhibitors in development will further elucidate the underlying biology and uncover its prognostic potential in this setting.”

Now: “In conclusion, this study extends the basis for understanding CLL pathogenesis and pathway dependencies that may be targeted by novel compounds. Identified molecular targets in a defined biologic context may further advance the development of new treatment strategies. Compound combinations targeting for example BCL2 and PRMT5 or XPO1, together with anti-CD20 monoclonal antibodies, may specifically synergize in genomically instable cases. Future assessment of the subtype related outcome in comprehensively characterized trial cohorts testing BCL2-, BTK- and other inhibitors in development will further elucidate the therapeutic potential of such treatment combinations.”

3. It is not clear that the clusters could not have been derived simply from DNA analyses: For example, for the GI/I-GI cluster it is stated that “cases with TP53 inactivation, V3-21 usage, short telomeres, high white blood cell (WBC) counts or ZAP-70 positivity ($p < 0.05$, Fisher’s exact test) were enriched in GI/(I)GI clusters. Furthermore, genes involved in maintenance of genomic stability were frequently mutated. Both GI and (I)GI involved frequent gains of 8q24.21 (including MYC) and 2p16.1 (including XPO1, REL). GI further showed gains of 6q22.31 and losses of 15q15.1 (including KNSTRN, BUB1B), 10q24.3 and 6q21. (I)GI showed losses of 13q14.13. Amplifications of MYC (8q24.21) were most frequently observed in GI/(I)GI cases. Moreover, IGHV mutated cases showed significantly higher activation of signatures 15 ($p = 0.01$), 3 and 20 ($p < 0.005$) (Mann-Whitney), indicating defective DNA repair.”

The authors should use the combination of the different DNA-based result to demonstrate that these are insufficient to generate precise clusters and that it requires the GEP in order to derive prognostically significant subgroups.

Response: We thank the reviewer for this important comment. To highlight this exact point and enhance understanding of our approach we generated a new figure S1G.

The figure illustrates that GEP is critical to the discovery of the identified pathogenic networks and subgroups. While well-known genetic associations (e.g. unmutated IGHV and short telomeres; unmutated IGHV and signature 9) were found as expected, the GEP cluster assignment could not be derived based on DNA-based markers alone.

Existing figure 3L was included to encompass parts of this and to extend understanding with regard to subtype specific biology, DNA-based variable inconsistency and the importance of GEP for categorization, but we agree that this could have been clearer and so include the new figure S1G.

Specific changes made to the manuscript:

We have conducted the proposed analysis of the DNA-based methods and included the respective figure S1G. NB: For this analysis we used only samples where all parameters have been assessed ($n = 162$).

Results, lines 171-173: “Optimal differentiation of distinct subtypes was achieved for k=6 GEP clusters (Fig.1B, S1E/F), while DNA-based class discovery approaches were insufficient to uncover similar patterns and the respective biological context (Fig.S1G).”

In addition, we have extended the survival analysis according to genetic alterations (PFS and OS) for cases from the GI, (I)GI, EMT-L and (I)EMT-L subtypes (Fig.5A-D, Fig.S7-S12, table S9). This analysis further highlights prognostic and biologic differences observed for GEP-based clusters (not detectable based on IGHV mutation status, or other DNA-based markers (see Fig.S1G, Fig.3L)), previously specified for genomic instability and AID activity in the results section also for Fig.3L.

The considerable prognostic impact of the GI subgroup is now better highlighted (poor clinical course of IGHV mutated cases) while parameters with strong prognostic impact were mostly balanced for IGHV mutated GI and (I)EMT-L cases (Table S9 + Table S2).

This analysis further supports the uncovered biologic and prognostic context, specifically observations made with regard to an increased genomic instability in cases with an increased activity of AID but insufficient DNA-repair as specified previously in results.

Specific changes made to the manuscript:

We have added the respective Fig.5A-D (Fig.5B showing PFS/OS with regard to IGHV) and table S8. Respective analysis was put into context in the results section lines 440-484.

Besides identifying prognostic categories, we have aimed to decipher pathogenic networks and interactions underlying CLL biology, for which GEP was essential. Information on pathway analyses, specific effects of deletions/amplifications translating into altered expression and specific evaluation of gene sets contained in the pathways or as deduced from GSEA (EMT, inflammatory signatures, methyltransferases, etc.) was derived from sequential analyses and with regard to clusters as identified through consensus clustering on GEP.

We identified and explained multiple network specific alterations which provide a novel perspective on disease biology.

An important aspect highlighting the importance of GEP data for accurate prognostic subgrouping in this study is provided by the fact that the major categories “genomically instable” and “epithelial-mesenchymal transition-like CLL” were heterogeneous regarding inflammatory and non-inflammatory subgroups.

Exemplary GI and (I)GI, which are highly similar with regard to high-risk (DNA-based) markers, were segregated from each other based on the inflammatory GEP signatures, but also in parts by GEP for DNA-repair genes (Fig.3A-D).

However, both clusters also differ considerably regarding other characteristics.

- Inflammatory subtypes (both (I)GI and (I)EMT-L) show lower lymphocyte counts compared to the non-inflammatory subtypes (Figure 1)
- (I)GI shows PFS similar to GI when treated with FC but a considerably better PFS when rituximab is added. (I)GI also shows the shortest OS (shorter than GI) for FC, but a better OS when rituximab is added. However, both subgroups can't be segregated when DNA-based classification is used (Fig.3L, Fig.S1G).

Specific changes made to the manuscript:

To specify that GEP based subtyping is essential regarding prognostic differences and differential response to treatment we have integrated Figure S7 as a main Figure, which together with Fig.S1G provides a better understanding of the results.

We further observe specific mutation enrichment patterns (e.g. EGR2, KRAS and other, Fig.S2B/C) suggesting a highly differentiated biology, which would not be identified with DNA-based approaches only (because of low frequencies for these subgroup specific gene mutations).

Fig.S1G now clearly highlights the necessity for GEP based subtyping regarding this aspect.

Specific associations of mutational signatures are impossible to deduct when DNA-based methods only are used (compare Fig.S1G). Especially with regard to differences for genomically unstable subtypes GI and (I)GI. Here, signature 2 is found preferentially in (I)GI and signature 6 is found preferentially in GI, indicating considerable biological differences (Fig.3J). However, these differences are only uncovered based on the clusters identified through consensus clustering and do not exclusively correlate with other parameters.

Fig.S1G now clearly highlights the necessity for GEP based subtyping regarding this aspect.

Clinical course of cases showing del13q, tri12 (according to the Döhner hierarchical model), or *SF3B1* mutations differs significantly when segregated based on the GEP based subtypes GI or (I)EMT-L (not otherwise achievable based on DNA-based markers as shown in Fig.1B / Fig.S1G).

To specify that GEP based subtyping is essential regarding prognostic differences and differential response to treatment we have added Fig.5A-D, Fig.S7-12 as main and suppl. figures, which now together with Fig.S1G provide a better understanding of the results.

4. The authors derive an immune signature that they then try to further validate in murine models. Unfortunately, I really cannot follow the result section relating to this and figure 4.

Please, include a rational for the choice of murine models and why precisely these models were used to either unpick the relationship between genomic instability and EMT stimuli and/or were chosen to model the specific disease context of CLL.

Please, summarise the experimental design and what you did at each step and why. This has to be summarised in each figure concisely and clearly. Please, focus on the key experiments only in the figure 4.

Response: We thank the reviewer for this important comment and apologize for being imprecise on describing the thinking process and reporting results in insufficient detail.

Four separate mouse models were chosen to validate observations on disease biology, respective pathogenic networks and dynamic processes *in vivo*.

We aimed to validate 4 specific aspects using these models

- 1) If inflammation can induce EMT-like changes in lymphoma *in vivo*
- 2) If genomic instability induced by MYC as a central driver in genomically instable CLL can inhibit EMT-like networks *in vivo*
- 3) If genomic instability induced by TCL1 as a central driver in genomically instable CLL can inhibit EMT-like networks *in vivo*
- 4) If EMT-plasticity is present in lymphoma and can be forced in aggressive tumors (with genomic instability) to a certain extent

1) We therefore aimed to validate the immediate effects of inflammation on EMT-induction in dynamic *in vivo* models. We have used two independent approaches (1+4) to confirm that continuous inflammation induces EMT-like changes.

The BCL1 tumor is a syngeneic lymphoma of BALB/c origin, originally described by Slavin and Strober. Inoculation results in a typical B-cell leukemia/lymphoma characterized by splenomegaly, peripheral blood lymphocytosis and death of all tumor-bearing mice. This model was used to show the induction of EMT as the BCL1 tumor develops over time; GEP changes indicating inflammation and EMT marker induction confirmed the continuous and strong inflammatory/immunological component and the corresponding EMT-like changes.

2+3) The next two mouse models (Eu-MYC and Eu-TCL1) were chosen to model genomic instability and validate inhibiting effects on EMT-like networks by utilizing a representative single oncogenic driver, which itself, or the respective pathway, has been identified as central element from the corresponding human data.

Along with multiple other alterations, we identified MYC pathway alterations as a core element of genomically instable CLL.

We also found multiple alterations leading to activation of the MYC-pathway, such as NMYC-amplifications, *IKZF3/IRF4* mutations and RAS/PI3K signaling or a loss of MYC repressors involving deletions of *MNT*, *MGA*, *PRDM1* in multiple cases with genomic instability. MYC-pathway activation was also confirmed thorough GSEA (Fig.2A) and western blotting (e.g. Fig.3H).

We therefore used the Eu-Myc model, developed by Adams et al., to validate effects of enforced MYC activation/overexpression. This is a syngeneic C57/BL6 model which places the c-Myc oncogene under the control of the immunoglobulin enhancer to induce lymphoid malignancy. This model provided a single driving oncogenic effect that facilitates downstream analysis of the result of its upregulation and was used to study whether genomic instability (and its consecutive effects) inhibits EMT networks.

We further identified that TCL1 overexpression is prominent in cases without inflammation and specifically genomically instable CLL. TCL1 is a major oncogenic driver in CLL and overexpression is further associated with inferior prognosis. We therefore hypothesized that TCL1 may contribute to genomic instability and to revert inflammation and EMT-like networks.

We therefore used the Eu-TCL1 model to assess effects of its activation/overexpression. Once again it is a spontaneous, syngeneic C57/BL6 cancer model, this time driven by the TCL1 oncogene under the control of the immunoglobulin enhancer, presenting over ~12 months with the gradual accumulation of tumor cells in the blood alongside splenomegaly and lymphadenopathy. It is considered by many to be the gold-standard in CLL mouse models and directly assesses the impact of TCL1 upregulation in B cells (mimicking the upregulation of TCL1 in CLL). It was therefore used to study whether genomic instability (and its consecutive effects) inhibits EMT networks in the context of a tumor expressing TCL1.

4) We further considered if repetitive forced induction of inflammation through serial transplantation could override TCL1-driven EMT-inhibiting effects and found that EMT-like induction can be forced upon such cells to a certain extent.

We have repeated the transplantation approach now using an E μ -TCL1 serial transplant mouse model with C57BL6/J recipient mice. This approach better reflects the previous conditions as we used the E μ -TCL1 transgenic model and use non-severely immunocompromised recipient mice with better potential for inflammatory response after inoculation. In addition we have increased sample size considerably and included the EMT-TFs *Zeb1* and *Snai1*.

Specific changes made to the manuscript:

We have considerably extended the relevant section and now provide extended explanations on the models used and the rationale. In addition we provide a better synthesis with the previous results sections and the overall content. Changes have been highlighted in red in the revised manuscript (lines 354-407).

We have repeated the TCL1 transplantation model with considerably more cases in a non-severely immunocompromised background. We have now included the transcription factors *Zeb1* and *Snai1* in this setting. Figures S5I/J/K have been adapted accordingly.

5. Regarding the epigenetic modification that might shape pathogenic networks and subgroupings: this claim is currently purely based on gene expression results. I would therefore advise to either remove the entire section or to produce supporting CHIP-Seq or ATAC-Seq data from a couple of key genes to show that key network genes are reprogrammed as a result of epigenetic modification.

Response: We thank the reviewer for this excellent comment. We have now extended the respective data and generated novel insights. We have conducted reduced representation bisulfite sequencing (RRBS) on available genomic DNA from n=182 CLL8 samples matching the cases with gene expression. For methylation analysis only CpGs with 10 or more reads were included. Promoters were defined as the regions encompassing 2 kb upstream and downstream of the transcription start site of UCSC genes. Promoters or genes with at least 5 covered CpGs were included into the analysis. Promoter or gene methylation was calculated by the average methylation levels of all the CpGs inside. Concrete methylation differences between groups were not observed when specifically evaluating single genes or differentially methylated genes in general and overall methylation levels. Using (I)EMT-L and GI (the two largest

groups) for comparison, we only identified 69 differentially methylated promoters out of 14559 investigated and 130 differentially methylated genes out of 15625 investigated ($p < 0.05$, methylation difference $> 5\%$, Mann-Whitney U -test). No promoters or genes were identified as differentially methylated after a Benjamini-Hochberg FDR procedure with BH-FDR $< 20\%$. Similarly, cases with *TP53* mutation and/or deletion did not exhibit differentiated methylation profiles. Interestingly we observed selective expression of DNA-demethylases in the (I)EMT-L subgroup, while AID/APOBECs and BER (associated with genomically instable cases) are known to be involved in demethylation processes. This data provides an important addition to understand the complex regulatory mechanisms underlying identified pathogenic networks and point to highly specific regulation of methylation/demethylation dynamics. We now provide reliable evidence that pathogenic networks, while initially unexpected, are not epigenetically pre-determined by methylation, rather that methylation seems narrowly regulated and that other epigenetic mechanism play a dominant role. Alternatively, respective epigenetic regulators may have additional roles in the network specific context. These findings considerably extend the understanding of our findings on CLL subtypes.

Beyond epigenetic regulation we have also taken a closer look at the EBF1-r cluster in this section of the manuscript (which we here used as an example to investigate epigenetic aspects, since it is transcriptionally highly distinct) and the finding of highly differentiated GEP in tri12 and some non-tri12 cases. We identify highly similar profiles for all tri12/ EBF1-r cases resembling healthy B cells, while epigenetic modifiers remain cluster specific.

For better clarity, we have now segregated the supplemental figures showing figures with regard to mouse experiments / EMT-induction only in Fig.S5 and all figures with regard to epigenetic modifiers/methylation in Fig.S6. We feel this clear segregation improves presentation of the results.

Specific changes made to the manuscript:

We have added the respective results in the results section (lines 429-435) and adapted supplementary figures S6, now showing methylation analyses for genes / promoters and the respective context (Fig.S6 J-k).

6. Cases without *TP53* defect showed PFS rates at 5 years of 17% in GI vs. 47% in (I)EMT-L (GI: median PFS 29.8 vs. (I)EMT-L: 39.5 months, HR:1.83 (95%CI 1.12-3.0), $p=0.016$) when treated with FC, but was the same for GI and EMT-L with the addition of rituximab.

In the CLL8 study, this shift of the PFS curve with the addition of Rituximab was also observed in patients with del11q. It is important to show whether the shift of GI reflects patients with these and/or other DDR abnormalities who are *TP53* wild-type, in particular also patients with bi-allelic *ATM* mutations. Conversely, in CLL8, a lack of benefit from the addition of rituximab was described for patients with *NOTCH1* abnormalities. These are enriched for in the EMT-L group that equally lacks this improvement.

More generally, it is important to understand that the prognostic relevance of the GI and EMT subgroups is independent and due to the specific GEP and NOT because of their association with *ATM* and other genomic abnormalities.

Related to this, PFS curves of the conventional markers TP53, IgHV, del11q, Notch1 and SF3BI should be shown as a comparator to investigate significant differences between these and the novel subgroupings based on GEP.

Response: We thank the reviewer for this important comment which helped to extend the analysis and better delineate prognostic associations and the underlying biology in identified subgroups. We fully agree with the raised points and are happy to provide additional confirmatory results with this analysis. We have introduced and explained the respective additions for survival analysis above (see response to comment #3), as these survival analyses also help to highlight the importance of using GEP for the identification of distinct biological subgroups.

We have added the extended analyses on the individual prognostic impact for recurrent genomic alterations with regard to its distribution across discovered subtypes in the manuscript as Fig.5A-D, we have added extended information on the analyses for survival times at 3, 5 and 7 years and median survival as well as numbers and events in Fig.S7-S.12. Survival differences, differential response to treatment and prognostic impact of recurrent genomic alterations or GEP based subgrouping are now much clearer.

Specific changes made to the manuscript:

We have included respective Figures, including extended information on survival times/PFS rates at 3, 5 and 7 years, median survival, numbers and events in the supplement.

Respective analyses were put into context in the results (lines 440-484):

Reviewer #2 (Remarks to the Author):

- Throughout figure 1, were all samples used, or just CLL8 samples. It looks like the initial characterization in Figure 1A might have been done in CLL8 and then validated with the REACH samples, but I am not sure if this is the case. If this is, were the EBF1 and NRIP1 clusters not found in the REACH samples?

Response: We thank the reviewer for raising this and apologize for imprecise presentation of the data.

Samples used for class discovery and detailed characterization of clusters were exclusively from the CLL8 cohort. For validation we have used the REACH cohort and an internal validation set from CLL8 (the latter not shown in the overview of identified subgroups in Figure 1A, but for better clarity in the CONSORT diagram).

While we could not identify clearly distinct clusters for EBF1-r and *NRIP1* in REACH, as was the case for GI, (I)GI, (I)EMT-L and EMT-L, we have validated the strong expression of EBF1-r specific signatures (shown in Figure Fig.S5N).

We have now also included an additional confirmatory figure for *NRIP1* expression with the inflammatory subtypes. Here, *NRIP1* was again specifically overexpressed in (I)GI and (I)EMT-L. We have adapted Figure 1A for the EBF1-r and *NRIP1* specific associations and specified this in the Figure legend.

Specific changes made to the manuscript:

Fig.1A: We have extended Fig.1A for better clarity, now indicating that EBF1-r signatures and *NRIP1* are validated in the respective biologic context for REACH but were not found as separate clusters. We have extended the figure legend specifying this aspect.

Fig.S.13E now shows the analysis of *NRIP1* expression with regard to major clusters (GI, (I)GI, (I)EMT-L and EMT-L) highlighting a specific overexpression in inflammatory clusters (I)GI and (I)EMT-L. Figure legend S13 has now been extended specifying this aspect.

- Was any validation of the GEP data performed on fresh cells? Both of the trials chosen have very old samples, and I think it would be important to make sure there is no freeze/thaw artifact. As well, were distinct differences found between the trials or between specific sites that initially processed the samples?

Response: We thank the reviewer for this important point and regret the lack of clarity in the original submission.

The samples were hybridized using cartridge arrays to be processed in relatively small numbers in parallel. Therefore, methodological batch effects would only be present in a small number of arrays unlike if we had used other systems like Gene Titan System where up to 96 samples can be processed in parallel. In addition, we have aimed for a high number of arrays/cases, so that effects of the date of the run would be balanced throughout sample groups. Labeling and hybridization of the arrays has been conducted consistently and in the shortest possible timeframe, at wintertime and under stable surrounding conditions.

We conducted an extensive assessment on the expression data for quality and with regard to potential batch effects imposed through e.g. time point of sampling, location of sampling, time point of labeling/hybridization and other factors and could not find any batches or associated impact on the data. For preprocessing we further used the "Robust Multichip Average (RMA)" algorithm providing resistance to outliers. In addition we conducted a quality control with "Relative Log Expression (RLE)" and "Normalized Unscaled Standard Errors (NUSE)" where we did not find any abnormalities. Notably, when reassessing distribution of cases across identified CLL subgroups with regard to potential batch inducing factors (time, age, location, etc.), we could not detect any imbalanced distribution or specific enrichment.

We have extended the information regarding these aspects in the supplementary methods part:

"We further assessed and excluded presence of potential batch effects induced by external factors including time point and location of sampling, duration of storage and time point of labeling and hybridization. Quality control was further conducted with "Relative Log Expression" (RLE) and "Normalized Unscaled Standard Errors" (NUSE), where abnormalities were not observed".

Importantly, we validated the identified subgroups through an independent cohort (the REACH trial).

- It would be helpful to see the specific genes that define the subgroups in figure 1b

Response:

Following class discovery, we subsequently defined biologic classes based on GSEA and with the assessment on differential expression for single genes/gene set, along with the applied genomics analyses. Gene sets represented in figures 1-5 and supplementary figures are all genes representing most differentially expressed between clusters as indicated. E.g. in Fig 3A-D we have provided significant FDRs of DEGs for GI vs. (I)EMT-L as indicated on the right (q) side of the figure.

We fully agree with the reviewer that early introduction of subtype defining genes is facilitating a comprehensible structure for the reader and increasing understanding of the logic deduction of results. We thank the reviewer for this comment which helps to increase the quality of the study.

Specific changes made to the manuscript:

We have extended the analysis on GEP for cluster defining genes. For this we assessed differential expression of genes in the cluster of interest against all other clusters. Both the top 10 up- and top 10 down regulated genes, fulfilling stringent cut-off criteria with a fold change ≥ 2 and $FDR < 0.0001$ were depicted as heatmap (corresponding CC k=6 clustering order) as Fig.S1F.

- Were the FISH, telomere length, TP53, zap-70, etc performed in a centralized manner for this study?

Response: All baseline parameters have been performed in a centralized manner in accredited reference laboratories of the German CLL Study Group (GCLLSG).

Specific changes made to the manuscript:

We have added this information to the supplementary methods as follows:

“All baseline parameters including genetics, serum parameters (such as thymidine kinase, β 2-microglobulin) and cell surface markers (such as ZAP-70) were performed in a centralized manner in accredited reference laboratories of the German CLL Study Group (GCLLSG) for the CLL8 trial, as outlined in the original study protocol. The central GCLLSG genetic reference testing laboratory in Ulm conducted fluorescence in situ hybridization (FISH), mutation analysis of genes recurrently mutated in CLL (such as *TP53*, *ATM*, *NOTCH1*, *SF3B1*) by targeted resequencing and IGHV mutation status, telomere length, GEP Exon- and SNP-Array hybridization and analysis.”

- It may be worth noting that the GI groups seemed much more likely to have TP53 mutation without del17p than the EMT groups (if this is actually significantly different; it does look so visually)

Response: We thank the reviewer for highlighting this important observation and are happy to bring this to the reader`s attention in a clearer way.

The observed differences for cases with *TP53* mutation without concurrent del17p were highly significant with:

p = 0.004 for the comparison GI vs. (I)EMT-L

p = 0.002 for the comparison GI and (I)GI vs. EMT-L and (I)EMT-L

Specific changes made to the manuscript:

We have added this information to the manuscript (lines 188-190) now specifying:

“*TP53* mutated cases without concomitant del17p showed a near-exclusive occurrence in genomically unstable cases (GI/(I)GI: n=16 (9.5%) vs. EMT-L/(I)EMT-L: n=1 (0.8%), p=0.002”.

- It is interesting that del11q frequency does not appear different among the clusters. It also looks like del11q appears more frequently than predicted (usually seen in about 20% of patients in frontline trials, visually looks like more than that).

Response: We thank the reviewer noting this and apologize for not being more precise in reporting the study details from the CLL8 trial. The CLL8 study (full cohort of n=817 patients) showed an incidence of 24.6% of cases with del11q. This incidence was similar to the later reported CLL10 trial (~24%). Patients with need for treatment enrolled on clinical trials may comprise populations with a higher incidence of del11q cases, as individual patients with more critical risk profile may be more likely to be enrolled on innovative trials due to the potential therapeutic improvement.

To provide further information on characteristics of patients from whom material was used for GEP, we have extended table S1. This table now provides information on patient characteristics for the full CLL8 trial cohort (n=817), the populations comprising

patients where no CD19+ sorted samples were available for GEP (n=480) and the CD19+ sorted GEP target analysis population (n=337).

We note a slight imbalance regarding del11q with 28.7% of cases in the CD19+ sorted GEP target analysis population (n=337) compared to the full cohort (n=817 patients); 24.6% del11q cases. However, when regarding *ATM* mutation and/or deletion, the difference was very small with 48.9% of cases in the CD19+ sorted GEP target analysis population (n=337) compared to the full cohort (n=817 patients); 50.1% of cases. In addition, del11q cases were equally distributed across major clusters ((I)EMT-L 28.3%, GI 31.8%, (I)GI 28.6%, EMT-L 26.7%), while pathogenic networks were retained irrespective of del11q.

Specific changes made to the manuscript:

We have added the information on frequencies of variables for the CD19+ sorted GEP target analysis population (n=337) in comparison to the full cohort (n=817 patients) in the table S1.

Do you think that 11q + another abnormality might be more important for GI or DDR abnormalities than del11q as a single abnormality?

Response: Cases with del11q were equally distributed across major clusters ((I)EMT-L 28.3%, GI 31.8%, (I)GI 28.6%, EMT-L 26.7%), while pathogenic networks (regarding distribution of genomic alterations, mutational signatures, GEP etc.) were retained irrespective of del11q. We analyzed the impact of del11q both with regard to the Döhner hierarchical model and sole presence of del11q and/or *ATM* mutations.

We found that del11q and/or *ATM* mutations itself did not show a heterogeneous outcome within the GI and (I)EMT-L subgroups, similar to cases with del17p and/or *TP53* mutations. This indicates that these lesions, themselves inducing genomic instability, may dominate over other identified pathogenic processes contributing to genomic instability.

However, outcome was highly heterogeneous with regard to the association with GI compared to (I)EMT-L for cases not exhibiting del11q and/or *ATM* mutations or del17p and/or *TP53* mutations, with cases falling into the GI category showing a much shorter PFS when treated with FC. These findings provide additional confirmation of our biological observations. However, when further extending subgroup analysis for multiple co-occurring variables, sample size was too low to provide reliable information on outcome.

Specific changes made to the manuscript:

We have added/extended the respective analyses for prognostic impact in subgroups and with regard to prognostic variables (exemplary KM-Plots shown below) (Fig. 5A-D, Fig.S7-12 and respective section in the manuscript (lines 440-484)).

- Since prognostically del13q is more relevant as a positive biomarker when it exists as a sole abnormality, I wonder if any of the analyses of chromosome abnormalities were performed separating out samples with del13q alone?

Response: We have added an extended analysis on the individual prognostic impact of recurrent genomic alterations with regard to its distribution across discovered subtypes as outlined above and in the revised manuscript.

We could show (Fig.S9) that (based on cytogenetics according to the Döhner hierarchical model), cases belonging to the (I)EMT-L subtype with del13, tri12, and cases without chromosomal aberrations show a better PFS than cases from the GI subtype when treated with FC. Similarly, this finding is reflected in Fig.5C/D where we extended analysis also with regard to *ATM*, *TP53* and *SF3B1* mutation status (which is not reflected in the hierarchical model).

Again, when extending analysis for distinct, selected additional subgroups, sample size was often reduced too much to provide reliable information on outcome.

Specific changes made to the manuscript:

We have added/extended the respective analyses for prognostic impact in subgroups and with regard to prognostic variables (Fig. 5A-D, Fig.S7-12 and respective section in the manuscript (lines 440-484)).

- Figure 2B is a little confusing for me to interpret, as I don't understand what the numbers at the top of the individual panels mean (% of samples? Clonal fraction?), and why is the scale and numbers at the top different for each of the panels. I also don't understand the interpretation that myc abnormalities are seen more frequently in the GI groups, when visually the (I)EMT group also seems to have a line in the same area as Myc in the GI groups. Also, the (I) EMT cluster seems to have a number of chromosome gains that would be interesting to know what those areas represent.

Response: We thank the reviewer for this comment and apologize for the lack of clarity regarding the description of the figure and results.

Figure 2B shows the results of the analysis for "Genomic Identification of significant targets in cancer" (GISTIC) using respective SNP-arrays. GISTIC identifies regions of the genome that are significantly amplified or deleted across a set of samples. Each aberration is assigned a G-score that considers the amplitude of the aberration as well as the frequency of its occurrence across samples. FDR q-values are then calculated for the aberrant regions, and regions with q-values below a defined threshold are considered significant.

In the provided figure the following findings are depicted: Chromosomes are oriented vertically from top to bottom (starting with chromosome1). GISTIC q-values at each locus are plotted from left to right on a log scale (bottom). GISTIC G-Scores (Frequency x Amplitude) are plotted on top of the plots. The green line represents the significance threshold (q-value = 0.25). Regions not reaching significance, as is the case for all peaks indicating gains in the (I)EMT-L cluster, were not evaluated or specified.

Specific changes made to the manuscript:

We have specified our description in the figure legend now saying:

"B) GISTIC analysis of copy number alterations. Chromosomal positions (1-22) on the y-axis indicate losses (blue, upper panels) or gains (red, lower panels) for major clusters. Affected genes representing CNA targets within biological networks (such as

YAP1) are shown for respective peaks. Most significant chromosomal peaks for major clusters are indicated on the right of each panel. GISTIC q-values at each locus are plotted from left to right on a log scale (bottom of each panel). Altered regions with FDR $q \leq 0.25$ (vertical green line) are considered significant. GISTIC G-Scores (amplitude of the aberration x frequency of its occurrence across samples) are plotted on top of the panels.”

- In Figure 3, I am not sure what the biologic relevance of total p53 or phospho p53 being elevated in the GI groups. While they are different, by itself I don't know that these basal levels are functionally relevant. Perhaps it would be of greater interest to show p53 induction after ionizing radiation.

Response: We thank the reviewer for this comment and apologize for the lack of clarity and being imprecise regarding the description of the figure and results.

For GI and (I)GI subgroups we have delineated multiple layers contributing to genomic instability. While *del17p/TP53* mutation is frequently seen as an inactivating event and therefore inducing genomic instability through diminished or missing DNA-damage response and repair, we observed that cases with *TP53* alterations cluster together with GI/(I)GI cases not exhibiting such alterations. Notably, our findings for the GI/(I)GI cases show an upregulation of the DNA-damage response and repair genes indicating that we here see an overactive but imprecise DNA-damage response and DNA-repair process. While genomic alterations provide a window towards events occurring at a given time point in the past and may impact future processes, we provide confirmation using GEP and especially the protein/phospho-protein data that the process is ongoing and much more activated in genomically instable CLL in comparison to EMT-L/(I)EMT-L. P53 upregulation and phosphorylation of p53 confirm a specific p53 activation in such cases. Notably, for the protein data we have used cases with wild-type status for critical genes like *TP53*, *ATM*, *MYC*, etc. to confirm that cases in the GI/(I)GI exhibit genomic instability and DNA-damage response activation irrespective of these alterations. We agree that irradiation induced upregulation would add information, but unfortunately we are unable to add data from primary (viable) samples. Cell line based irradiation data is already present (Figure S5H) where we could confirm active induction of *TP53*.

Specific changes made to the manuscript:

We have extended our description of these findings in the manuscript (lines 246-249) and figure legend now saying: “Importantly, upregulation of p53 and phospho-p53 protein levels was confirmed in genomically instable cases without recurrent gene mutations or chromosomal aberrations other than *del(13q)* (Fig.3E) and confirm a continuous activation independent of such lesions.”

- Why do the survival curves in Figure 5 only include (I) EMT and not the non-inflammatory cluster?

Response: In this figure we initially aimed to depict survival differences and treatment impact arising from the specific biology (EMT-like networks or genomically instable, FC vs. FCR) in major CLL subgroups, independent of *TP53* mutations/deletions. *TP53* *del/mut* cases were therefore segregated as separate curve. Together with figures provided for the REACH cohort, following the identical scheme, we aimed to show that

in the relapse situation *TP53* wild-type cases falling into the G1 cluster perform equally poorly as *TP53* mut/del cases.

Extending survival analysis for subgroups (also see comments above), we have now included the analysis on all subtypes for both treatment arms along with the other panels showing specific subgroup analyses.

Specific changes made to the manuscript:

We now have included Fig.S.7A PFS and OS KM-Plots as a main figure, together with the extended survival analysis for other prognostic markers in the newly generated main Fig.5. We have kept Fig S.7 unchanged to also provide extended information on survival times at 3, 5 and 7 years, median PFS, detailed information for patient numbers in treatment arms and events.

REVIEWER COMMENTS

Reviewer #1 (Remarks to the Author):

the authors have done a very comprehensive revision and answered my concerns. I am happy with the revised version.

Reviewer #2 (Remarks to the Author):

The authors have done a nice job of responding to reviewer comments and I have no further comments.